# Development and Application of a Senolytic Predictor for Discovery of Novel Senolytic Compounds and Herbs

**DOI:** 10.3390/molecules30122653

**Published:** 2025-06-19

**Authors:** Jinjun Li, Kai Zhao, Guotai Yang, Haohao Lv, Renxin Zhang, Shuhan Li, Zhiyuan Chen, Min Xu, Naixue Yang, Shaoxing Dai

**Affiliations:** 1State Key Laboratory of Primate Biomedical Research, Institute of Primate Translational Medicine, Kunming University of Science and Technology, Kunming 650500, China; 20222140064@stu.kust.edu.cn (J.L.); lhhmail508@163.com (H.L.); zhangrenxin1999@163.com (R.Z.); lishuhan1@stu.kust.edu.cn (S.L.); chenzhiyuan_7715@163.com (Z.C.); 2Yunnan Key Laboratory of Primate Biomedical Research, Kunming 650500, China; 3Center for Pharmaceutical Sciences, Faculty of Life Science and Technology, Kunming University of Science and Technology, Kunming 650500, China; zk225dizzy@126.com (K.Z.); yangguotai@kust.edu.cn (G.Y.)

**Keywords:** senolytic compounds, machine learning, virtual screening, MoLFormer, DrugBank, TCMbank

## Abstract

The accumulation of senescent cells is a major contributor to aging and various age-related diseases, making developing senolytic compounds that are capable of clearing these cells an important area of research. However, progress has been hampered by the limited number of known senolytics and the incomplete understanding of their mechanisms. This study presents a powerful senolytic predictor built using phenotypic data and machine learning techniques to identify compounds with potential senolytic activity. A comprehensive training dataset consisting of 111 positive and 3951 negative compounds was curated from the literature. The dataset was used to train machine learning models, incorporating traditional molecular fingerprints, molecular descriptors, and MoLFormer molecular embeddings. By applying MoLFormer-based oversampling and testing different algorithms, it was found that the Support Vector Machine (SVM) and Multilayer Perceptron (MLP) models with MoLFormer embeddings exhibited the best performance, achieving Area Under the Curve (AUC) scores of 0.998 and 0.997, and F1 scores of 0.948 and 0.941, respectively. This senolytic predictor was then used to perform virtual screening of compounds from the DrugBank and TCMbank databases. In the DrugBank database, 98 structurally novel candidate compounds with potential senolytic activity were identified. For TCMbank, 714 potential senolytic compounds were predicted and 81 medicinal herbs with possible senolytic properties were identified. Moreover, pathway enrichment analysis revealed key targets and potential mechanisms underlying senolytic activity. In an experimental screening of predicted compounds, panaxatriol was found to exhibit senolytic activity on the etoposide-induced senescence of the IMR-90 cell line. Additionally, voclosporin was found to extend the lifespan of *C. elegans* more effectively than metformin, demonstrating the value of our model for drug repurposing. This study not only provides an efficient framework for discovering novel senolytic agents, but also highlights the predicted novel senolytic compounds and herbs as valuable starting points for future research into senolytic drug development.

## 1. Introduction

Aging is a complex process, and twelve hallmarks of aging have been identified, including genomic instability, epigenetic alterations, mitochondrial dysfunction, and cellular senescence, among others [1,2]. Among these hallmarks, cellular senescence is considered a key driver of aging and age-related diseases. Senescent cells are characterized by stable cell cycle exit and loss of proliferative capacity, entering a state of permanent growth arrest [3]. These cells exhibit distinct molecular and morphological features and are typically associated with the secretion of varied bioactive factors, including inflammatory cytokines, chemokines, growth factors, matrix metalloproteinases, etc., known as the senescence-associated secretory phenotypes (SASPs) [4].

With advancing age, the burden of senescent cells increases significantly in various tissues and organs, such as brain, lung, liver, adipose tissue, blood vessels, skin, and bone, exerting multiple negative effects on the organism [5,6,7,8,9,10]. The persistent SASP induced by senescent cells enables them to resist clearance and may trigger chronic inflammation, leading to age-related diseases and even cancer. Studies have shown an inverse relationship between senescent cell burden and a healthy lifespan in mice [11]. The selective removal of p16 high-expressing cells from progeroid mice improved their healthspan, extended the median lifespan, and delayed tumorigenesis [12]. However, transplanting a small number of senescent cells into young, healthy mice resulted in age-related physical dysfunction [13,14]. These findings highlight the negative impact of senescent cell accumulation on health and suggest that reducing the burden of senescent cells could extend healthspan and alleviate age-related diseases.

Therefore, senescent cells have become a promising therapeutic target. One of the most promising drug types for targeting senescent cells are senolytics, which alleviate senescence by targeting and eliminating senescent cells. These drugs lend themselves to a “hit-and-run” delivery strategy, which can effectively limit the negative effects of senolytics due to off-targeting or interference with beneficial cell populations [15].

Senolytics were first discovered and defined in 2015 [16]. Zhu et al. identified the activity of dasatinib (D) and quercetin (Q) as senolytics from a screen of anti-apoptotic pathway modulators. After nearly a decade of research development, the senescent cell scavenger activities of drugs such as Bcl-2 family protein inhibitors, cardiac glycosides, P53 stabilizers, and BET family protein inhibitors have been identified [17]. However, many senolytics are known to have side effects, such as thrombocytopenia caused by ABT-263 [18], and cardiac glycoside analogs, which often caused an abnormal heart rate and blood pressure, gastrointestinal and nervous system side effects, etc. [19]. In addition, many senolytics are only effective against specific cell types or senescent cells resulting from specific senescence-inducing conditions [20,21], so while senolytics may be suitable for a specific indication, they also limit broader clinical applications. Currently, the number of senolytics is small and clinical trials are still largely limited to D + Q and fisetin, with few advancing to phase 3 or beyond [22]. Therefore, developing novel senolytic drugs is of great practical importance and application value.

Artificial intelligence-based computational screening has been widely adopted in industry and academic research over the past decade [23]. Some of these strategies differ from traditional target-based drug discovery approaches, in that models are trained directly from phenotypic data results, a strategy that does not require knowledge of the drug target [24]. This target-free approach provides new ways to expand the chemical starting point in the early stages of the drug discovery process [25], and is particularly applicable to the search for anti-aging drugs, as our current understanding of the molecular pathways controlling the cellular senescence phenotype is still relatively limited. Recently, both Wong et al. [26] and Smer-Barreto et al. [27] used human lung fibroblasts to construct a model of senescent cells that was combined with a target-free machine learning model to successfully screen for senolytics. However, due to the limited number of senolytics and the heterogeneity of these drugs, the performance of both models remains suboptimal. For instance, the model proposed by Wong et al., achieved an auPRC (Area Under the Precision–Recall Curve) of just 0.243 [26], while the model developed by Smer-Barreto et al. attained an F1 score of only 0.276 [27].

To address the existing limitations in senolytic discovery, this study aimed to develop an improved senolytic predictor through systematic methodology. Known senolytics include many types of biological activity, such as inhibitors of various kinases, anticancer drugs, and cardiac glycosides, among others. Given this heterogeneity and the relatively small number of senolytics, a phenotype-based strategy was used in this study to construct predictive models. Building upon previous work by Wong et al. [26] and Smer-Barreto et al. [27], their findings were integrated with additional screening data to compile an active training set of known senolytics. The methodology employed a rigorous comparison of diverse molecular feature representations and machine learning architectures to identify optimal modeling approaches. This computational framework facilitated screening both the DrugBank dataset for potential drug repurposing candidates and the TCMbank database to explore natural products with senolytic potential. Subsequent enrichment analysis enabled inferring medicinal herbs potentially harboring senolytic compounds. The preliminary experimental validation of selected candidates was incorporated to assess the framework’s utility. Collectively, this approach established a refined methodology for identifying senolytics through computational prediction and experimental prioritization.

## 2. Results

### 2.1. Workflow for Developing Senolytic Predictor

The development process of the senolytic predictor is illustrated in Figure 1. Due to the limited number of currently known senolytics, compounds identified as senolytics were collected from the published literature [26,27,28,29,30,31,32,33,34,35,36,37,38,39,40,41,42,43,44,45,46,47,48,49,50,51,52,53,54,55]. These compounds were validated for activity in at least one cell line and could preferentially clear senescent cells. For the negative dataset, previous senolytics screening efforts were integrated [26,27,28,56], which included compounds experimentally validated as lacking senolytic activity and other compounds used effectively as inactive datasets in previous virtual screening studies. To ensure the reliability of the negative dataset, we compared the molecular fingerprint similarity between compounds in the negative and positive datasets and excluded those with more than 90% similarity to known senolytics. Finally, the training set contained 111 positive compounds and 3951 negative compounds. Subsequently, the training set was used to build machine learning models. By comparing the performance of different molecular features and machine learning models, the best-performing SVM and MLP were selected to assemble the senolytic predictor. The t-distributed stochastic neighbor embedding (t-SNE) was employed to visualize the distribution of training set compounds and query compounds from DrugBank and TCMbank in chemical space (Appendix A). The training set of 4062 compounds sufficiently occupied the majority of the chemical space of tens of thousands of query compounds, indicating that the predictor could make relatively reliable predictions for these query compounds. We then used the predictor for virtual screening to discover potential senolytics.

### 2.2. SVM and MLP Models with MoLFormer Molecular Embeddings Perform Best

Given the scarcity of positive data and the severe imbalance of the training set, we tested different molecular feature representations and machine learning models to construct the senolytic predictor. For molecular feature representation, we used traditional methods such as FP2 (Open Babel Fingerprints 2) molecular fingerprints, ECFP (Extended Connectivity Fingerprints) molecular fingerprints, and molecular descriptors, which have been proven to be effective in molecular activity prediction [57]. Additionally, we used molecular embeddings generated using the pre-trained neural network model MoLFormer [58], which uses rotational position embeddings and linear attention mechanisms to train on SMILES (Simplified Molecular Input Line Entry System) sequences of 1.1 billion unlabeled molecules from the PubChem and ZINC databases; this outperformed traditional molecular representations in some downstream tasks. For machine learning models, we experimented with Random Forest (RF), Support Vector Machine (SVM), eXtreme Gradient Boosting (XGBoost), Convolutional Neural Networks (CNN), and Multilayer Perceptron (MLP). After hyperparameter tuning, the overall performances of the models were quantified using 20 iterations of repeated random-split cross-validation, averaging multiple classification evaluation metrics, including accuracy, AUC, precision, recall, and F1 score (Figure 2a–e). Limited by the number and heterogeneity of positive data, all models had a recall below 0.5, but models using MoLFormer molecular embeddings generally had better recall. Overall, SVM and MLP models using MoLFormer molecular embeddings achieved the best performance (Figure 2b,d), with precision scores both above 0.78, AUCs of 0.888 and 0.875, and F1 scores of 0.504 and 0.502, significantly outperforming other models. Consistent results were also shown for evaluating confusion matrices [59] and the MCC (Matthews Correlation Coefficient) [60] (Appendix A). Therefore, we then constructed the senolytic predictor using SVM and MLP models with MoLFormer molecular embeddings as inputs.

### 2.3. Enhanced Senolytic Predictor Through Optimized Machine Learning Models and Oversampling Strategies

After hyperparameter optimization, the SVM model achieved an AUC of 0.916 and an auPRC of 0.685 (Figure 3a,c). In comparison, the MLP model reached an AUC of 0.902 and an auPRC of 0.639 (Figure 3b,d). To further improve the accuracy and stability of the MLP model, we used an ensemble approach by generating 10 sub-models to obtain a comprehensive prediction result [61] (Appendix A). The performances of the models were compared to those of previous senolytic prediction models. The SVM and MLP models showed significant improvements in the recall and F1 score compared to the XGBoost model of Smer Barreto et al. [27] (Figure 3f). Additionally, the model’s auPRC showed a significant increase compared to the MPNN model of Wong et al. [26] (Figure 3c–e). Overall, acquiring more senolytics for the training set predictably improved the model’s performance.

To effectively address the challenge of class imbalance in our dataset, we utilized the unique properties of the MoLformer model to oversample positive samples. The MoLformer model generated different molecular embeddings for the same molecule when processed in different batches, which allowed us to enhance data diversity and representation during oversampling. Specifically, a 5-fold oversampling strategy was applied to the dataset, as described in the Methods section. This strategy proved to be highly effective in balancing the dataset. Despite the variation in molecular embeddings, the active compounds present in the test set were potentially included in the training set. As a result, the recall of the model improved substantially, reaching nearly 1.0 (Figure 3g). More importantly, this oversampling approach also significantly improved model precision, reducing the rate of false-positive predictions (Appendix A). Consequently, this oversampling strategy formed the basis for constructing the senolytic predictor. Building on this strategy, a robust senolytic predictor was developed that integrated both the SVM and MLP models. The SVM and MLP models achieved AUC scores of 0.998 and 0.997, F1 scores of 0.948 and 0.941, and MCC scores of 0.970 and 0.970, respectively (Figure 3g, Appendix A). The two models operate simultaneously to predict senolytic activity. To further ensure the reliability of predictions, only compounds classified as positive by both models were considered potential senolytics. This dual-model approach not only ensured robustness in prediction, but also minimized false-positives, making it a highly reliable senolytic prediction system.

### 2.4. Discovery of New Senolytics Through DrugBank Dataset Prediction

The DrugBank database is a comprehensive resource containing detailed information on a wide variety of compounds, including approved drugs and experimental drugs [62]. To discover potential senolytic agents, senolytic activity predictions were performed on small-molecule drugs in the DrugBank database. The senolytic predictor identified a significant number of potential senolytics, including the following: the MLP model predicted 275 candidates, while the SVM model predicted 231, and all of these predicted active compounds were within the AD, indicating that the predicted results for these compounds were relatively reliable (Appendix A). Among these potential active compounds, some known senolytics were identified via a comparison with active compounds in our training set (Figure 4a). Special focus was placed on compounds receiving positive predictions from both models. There were 130 in total (Figure 4b). Furthermore, by comparing the molecular fingerprint similarity of these compounds to known senolytics in the training set, most of these compounds were structurally novel, with 98 compounds having a structural similarity below 0.5 and 81 compounds having one below 0.3, compared to known senolytics (Figure 4c). This result is consistent with the t-SNE analysis using MolFormer molecular embeddings (Figure 4d), suggesting that the senolytic predictor can partially recognize specific biological effects arising from different compound substructures, thereby predicting structurally novel compounds.

The activity information for these top 10 compounds was further analyzed (Table 1). Among them, cymarin, metildigoxin, acetyldigitoxin, and acetyldigitoxin belonged to the class of cardiac glycosides, which have been previously identified as a class of senolytics [55], and several cardiac glycosides, including digitoxin, were also identified as senolytics. Amrubicin and Aclarubicin are topoisomerase II inhibitors, as is the known senolytic idarubicin. Quisinostat is an HDAC inhibitor, and HDAC inhibitors have been reported to be senolytic [44]. Notably, the targets of several other compounds were found to be relatively novel when compared to known senolytics, suggesting the potential discovery of new mechanisms of action. Overall, these results suggest that the predictions of our senolytic predictor have a certain reference value and warrant further experimental validation to determine the true activity of these compounds.

### 2.5. Identification of Structurally Novel Senolytic Candidates from the TCMbank Database

TCMbank is currently the largest database of traditional Chinese medicine (TCM), containing extensive information on active molecules from TCM [63]. By utilizing this rich resource, we applied our senolytic predictor to predict senolytic activity among small molecules in the TCMbank database. The MLP model identified 933 potential senolytics, while the SVM model identified 1170 potential senolytics, and all of these predicted active compounds were within the AD, indicating that the predicted results for these compounds were relatively reliable (Appendix A). Among these, 714 compounds were simultaneously predicted as senolytics using both models (Figure 5a,b), making them particularly promising candidates for further analysis. To evaluate the structural uniqueness of these 714 compounds, a structural similarity assessment was evaluated. Consistent with findings from the DrugBank predictions, most hits exhibited structural similarity scores concentrated in the range of 0.2–0.3, indicating that these compounds are not only structurally novel, but also diverse and complex (Figure 5c,d). The top 10 ranked compounds are shown in Table 2. Among these, bufotalidin, helveticoside, cymarin, cymarol, and malayoside were identified as cardiac glycosides, a class previously associated with senolytic activity [54,55]. However, less is known about the biological activity of the other compounds, highlighting the need for further exploration.

### 2.6. Identification of Potential Senolytic Medicinal Herbs

Many hundreds of compounds with potential senolytic activity were predicted. It was worth identifying the medicinal herbs that are enriched with senolytic compounds. Identifying senolytic herbs is of great value for the introducing, utilizing, and protecting medicinal herbs. It is also important for developing senolytic drugs. Therefore, based on the predicted senolytic compounds, an enrichment analysis was subsequently performed, mapping the 714 predicted active compounds to their corresponding medicinal herbs. This analysis resulted in the identification of 81 enriched medicinal herbs (P_adj  <  0.05) (Appendix A), of which the top 20 are shown in Table 3. The top five enriched medicinal herbs with the highest enrichment scores were *Bufo bufo gargarizans*, *Erysimum cheiranthoides*, *Strophanthus divaricatus*, *Thevetia neriifolia*, and *Corchorus capsularis*, which may be potential senolytic medicinal herbs. To verify this result, a literature survey was conducted using the PubMed database. Some compounds from these herbs have been identified as senolytics, such as bufalin from *Bufo bufo gargarizans* [55], and oleandrin from *Nerium indicum* [27]. In addition, extracts of some herbs have shown anti-aging activity, such as *Morus Alba* L. [64] and *Bupleurum smithii* [65]. An analysis of herb classifications revealed a distribution across 46 genera and 29 families. An examination of family distributions indicated elevated frequencies of senolytic herbs in the families of Apocynaceae, Selaginellaceae, Ranunculaceae, Apiaceae, and Brassicaceae. For example, there are 14 senolytic herbs belonging to the Apocynaceae family, which are as follows: *Strophanthus divaricatus*, *Thevetia neriifolia*, *Strophanthus kombe*, *Tabernaemontana corymbosa*, *Cerbera manghas*, *Cerbera odollam*, *Nerium indicum*, *Beaumontia grandiflora*, *Periploca nigrescens*, *Asclepias curassavica*, *Marsdenia koi*, *Ervatamia divaricata*, *Asclepias eriocarpa*, and *Nerium oleander*. These newly identified senolytic herbs are worth studying further to provide more opportunities for developing senolytic drugs.

### 2.7. Target Analysis and Pathway Enrichment Reveal Potential Mechanisms of Senolytic Activity

A subsequent examination of the target information of these potential active compounds revealed that most of them lacked clear target records in TCMbank (Appendix A), suggesting that research on these compounds is still insufficient. Statistical analysis of available target data identified 343 unique gene targets corresponding to these compounds. A comparison of these compounds to known senolytics revealed that they share 74 common targets (Appendix A), including the well-known senolytic targets BCL2, MCL1, PPARA, and TP53 [20,32,36,39]. Targets with more than 10 occurrences are shown in Figure 5e, and most of them are also targets of known senolytics. These genes play key roles in various biological processes, including apoptosis, autophagy, proliferation, and immune regulation. For example, ESR1 and AR are crucial in hormone-regulated cell proliferation, which affects the cellular microenvironment through inflammatory responses. Moreover, PTGS2 and PIM1 play important roles in regulating cellular senescence, promoting cell survival by inhibiting apoptosis [66,67]. In summary, these frequently hit gene targets may represent potential therapeutic targets for senolytics (Figure 5e).

Further KEGG (Kyoto Encyclopedia of Genes and Genomes) enrichment analysis [68] of these target genes showed an interesting result, with the neuroactive ligand–receptor interaction pathway scoring the highest, possibly because our predicted results contained many compounds with modulated neurotransmitter or enhanced neuroactive ligand activity (Figure 5f). In addition, many cancer-related pathways were enriched because many known senolytics and predicted compounds also have anticancer activity. Other pathways were mainly related to infectious diseases, consistent with the established relationship between aging and immune system alterations, including weakened immune function and increased chronic inflammation. Senescent cells influence the microenvironment by secreting SASP. These changes may be associated with increased susceptibility to infectious diseases.

### 2.8. Cellular Assay for Senolytic Activity of Candidate Compounds

To facilitate the subsequent evaluation of predictive results, additional predictions on compounds from the FDA_HY-L022 and TCMM_HY-L065 databases were conducted, as these compounds are readily available. The FDA_HY-L022 database comprises compounds that have been approved by the FDA for marketing, while the TCMM_HY-L065 database includes naturally occurring bioactive molecules derived from traditional Chinese medicine. By merging the 2485 compounds from the FDA_HY-L022 database with the 1771 compounds from the TCMM_HY-L065 database, and removing duplicates, we ultimately obtained a query dataset containing 4023 unique compounds (Appendix A). By employing a drug repurposing strategy, the activity predictions on the compounds within these two datasets were performed to discover novel senolytics. The senolytic predictor identified several potential senolytic candidates, as follows: the MLP model predicted 115 candidate compounds, whereas the SVM model predicted 122 candidates. A comparative structural analysis between these candidate compounds and active compounds in the training set revealed 47 known senolytic agents (Figure 6a,b). Furthermore, 47 compounds receiving positive predictions from both SVM and MLP models were designated candidate senolytics. Similar to the predicted results for Drugbank and TCMbank, most of the candidate compounds have novel chemical structures compared to known senolytics (Figure 6c,d).

One of the side effects of chemotherapy in cancer treatment is that the cytotoxicity of chemotherapeutic agents can induce normal somatic cells to exit the cell cycle and enter a state of senescence. The IMR-90 cell line is one of the most commonly used models for screening senolytic drugs [17]. In this study, etoposide was employed at a concentration of 50 μM to induce senescence in IMR-90 cells. Following treatment, the cells exhibited pronounced senescence, as confirmed via the detection of the senescence-associated β-galactosidase (SA-β-gal) marker (Figure 6e). ABT-263, a known Bcl-2 inhibitor, has binding activity against several members of the Bcl-2 protein family, including Bcl-2, Bcl-w, and Bcl-xL. In 2016, Zhu et al. identified ABT-263 as a senolytic agent, with Bcl-2 family proteins serving as its functional targets [69]. Accordingly, ABT-263 was used as a positive control in our experiments. In validation studies (Figure 6f), treatment with 10 μM ABT-263 resulted in approximately 50% elimination of senescent cells, while about 80% of proliferating cells were preserved. These results are consistent with those reported by Zhu et al. [69].

By using the chemotherapy-induced senescent IMR-90 cell model, the senolytic activity of the six compounds selected from the forty-seven predicted candidates were evaluated. Additionally, four compounds predicted to be inactive were chosen as negative controls. The control group received 0.1% DMSO, while the positive control was treated with 10 μM ABT-263. The experimental groups were treated with various concentrations of the test compounds. For panaxatriol, preliminary experiments indicated that 10 μM was capable of killing approximately 10% of senescent cells without toxicity to normal cells; therefore, its activity was further examined at both 10 μM and 100 μM. The other compounds were tested at concentrations of 1 μM and 10 μM. As shown in Figure 6g, the positive control of ABT-263 displayed typical senolytic activity by preferentially eliminating senescent cells. Among the 10 compounds tested, panaxatriol demonstrated notable senolytic activity at 100 μM, clearing about 45% of senescent cells, while only approximately 12% of the proliferating cells were eliminated. The remaining compounds did not exhibit senolytic activity at either 1 μM or 10 μM concentrations.

### 2.9. The C. elegans Lifespan Assay Reveals the Anti-Aging Effects of the Candidate Compound Voclosporin

At the same time, experimental screening of model-predicted candidate compounds was recently initiated using the *C. elegans* model. The experimental study is still at an early stage, so only a few results are currently available. Some of the current results are presented in Table 4 and Figure 7. As senolytics belong to a class of anti-aging drugs, we are more interested in drugs that improve the lifespan or healthy lifespan of an organism. Therefore, *C. elegans* was chosen as the model for the preliminary experimental screening. *C. elegans* is a commonly used model animal in anti-aging research, and compared to in vitro cellular models, the *C. elegans* model can better assess the physiological effects and anti-aging activity of drugs at the whole-organism level. As shown in Table 4, the predicted active compound voclosporin markedly extended the lifespan of *C. elegans*, achieving a lifespan extension rate of 19.1%, which is better than that of the well-known anti-aging drug metformin (11.3%). Although voclosporin did not extend the lifespan of the longest-lived *C. elegans*, it significantly reduced *C. elegans* mortality at the onset of senescence and at around 30 days (Figure 7). The other four compounds (simeprevir, belinostat, paritaprevir, and tenapanor), which were predicted to be inactive, had no effect on extending the lifespan of *C. elegans*. Voclosporin, from the DrugBank dataset, is a known calcineurin inhibitor used for treating autoimmune diseases [70]. Currently, no published anti-aging studies exist for voclosporin. The specific anti-aging mechanisms of voclosporin will require further experiments to elucidate.

## 3. Discussion

First of all, this study successfully developed a machine learning-based senolytic predictor using comprehensive phenotype-based data. Then, by performing virtual screening on the DrugBank and TCMbank datasets, several potential senolytics were identified. The results not only validate the potential of machine learning in drug discovery, but also offer new chemical starting points for developing senolytics [25].

In previous senolytic drug discovery efforts, researchers have screened for senolytics using different senescent cell models, including different cell types such as IMR-90, HUVEC, and A549, among others, as well as different senescence strategies, such as replicative senescence and chemotherapeutic drug (e.g., adriamycin and etoposide)-induced cell senescence. The senolytics identified through these screening efforts also have diverse targets and mechanisms, including BCL2 family protein inhibitors, BET family protein inhibitors, and P53 stabilizers, among others. Moreover, due to the small number of senolytics discovered so far, these heterogeneities are difficult to discuss, highlighting the value of target-agnostic machine learning strategies. Therefore, a phenotype-based strategy was used in this study to construct the prediction model.

A key finding of this study is that increasing the size of the positive dataset improves the performance of the senolytic predictor, even with the inherent heterogeneity of the compounds. Dataset expansion incorporating novel senolytics from the literature nearly doubled the size of the previous datasets used in comparable studies, such as those by Wong et al. [26] and Smer-Barreto et al. [27]. Despite the increase, the dataset remains relatively small, which limits further improvements in model performance. In addition, the heterogeneity of the compounds, due to different experimental conditions across different cell lines [20], limits the reliability of predictions and results in a relatively low recall (Figure 3f). Consequently, our model exhibits significant variability in predictions on external datasets (Figure 4b and Figure 5b). In the future, a larger and more homogeneous set of validated senolytics from consistent experimental conditions could improve the model’s performance, especially in terms of recall.

To mitigate issues related to sample imbalance and feature inconsistency due to the random initialization of different batches, we employed an innovative oversampling strategy based on the MoLFormer model. Unlike traditional oversampling techniques such as SMOTE, which generate synthetic data, our strategy generates real positive compounds with varied but similar molecular embeddings. This enhances the model’s ability to learn from these compounds more effectively (Figure 3g). However, this approach does carry a risk of overfitting, as repeated sampling may alter the distribution of samples in the feature space. To avoid this, we used 5-fold oversampling instead of equalizing the number of positive and negative samples. To ensure the validity of predictions, only compounds predicted to be positive using both the SVM and MLP models were considered as potential senolytics, thereby improving the reliability of our predictions.

The MoLFormer embeddings, in conjunction with the SVM and MLP models, performed exceptionally well in predicting senolytic activity, outperforming traditional models such as decision trees, which struggled with the high dimensionality of the MoLFormer embeddings (Figure 2a,c). This highlights the superior performance of MoLFormer in capturing the bioactive features of compounds and underscores the growing importance of deep learning models in cheminformatics, especially for complex biological datasets.

The ability of the predictor to predict structurally novel compounds was exemplified by screening the DrugBank dataset. To establish robust confidence thresholds, predicted compounds were ranked by dual-model probability scores (SVM > 0.0 and MLP > 0.5). Further, by comparing the structural similarities, 98 of these compounds showed significant differences from known senolytics, suggesting the potential of the model to predict novel bioactive compounds. This is particularly promising for drug repurposing in the senolytic field, where structurally novel compounds may exhibit better drug activity and fewer side effects.

The parallel screening of TCMbank employed identical confidence thresholds, identifying 714 potential senolytic compounds. Subsequent enrichment analysis pinpointed 81 medicinal herbs enriched in these candidates, providing systematic leads for natural product-derived senolytic development. In particular, all of the herbs of *Bufo bufo gargarizans*, *Erysimum cheiranthoides*, *Strophanthus divaricatus*, and *Thevetia neriifoliaall* contain varied cardiac glycosides and emerged as potential senolytic sources [71,72,73,74]. In addition, *Corchorus capsularis* has a wide range of biologically active constituents with pharmacological properties such as antitumor, antioxidant, and anti-inflammatory properties that help to prevent and treat many chronic diseases [75]. *Morus alba* L. also has a wide range of pharmacological activities, such as analgesic, anti-inflammatory, hypoglycemic, and antimicrobial [76]. These findings provide new insights into the development of natural products with senolytic properties and support the modernization of traditional Chinese medicine.

This study employs a two-model strategy, which will inevitably miss true-positives confidently predicted by only one model. However, for Cost–Benefit Justification reasons, the cost of lowering the recall by a certain amount to reduce false predictions is worthwhile. However, while the model’s predictions are promising, they should be considered preliminary references rather than definitive conclusions. Experimentally validating these predicted compounds is crucial to confirm their biological activity. Our next step will involve validating the senolytic activity of these compounds experimentally, which will also provide an opportunity to further refine our model based on experimental data. Another limitation of this study is the poor interpretability of the model’s results. Although the phenotypic data-driven approach is powerful for predicting senolytic activity, it does not provide insight into the underlying mechanisms or drug targets. To address this, molecular docking is planned to explore potential drug targets [77], although detailed experimental validation will be required for more reliable insights.

In total, six of the forty-seven candidate compounds have been evaluated for senolytic activity at the in vitro cellular level. In the etoposide-induced IMR-90 senescent cell model, panaxatriol exhibited senolytic activity at a concentration of 100 μM, killing nearly half of the senescent cells. Recently, it was reported that panaxatriol can target UFL1 to exert anti-aging effects, and can alleviate fibrosis in osteoarthritis and cartilage repair [78]. Na^+^, K^+^-ATPase is a known target of action of cardiac glycoside senolytics, and Wu et al. found that panaxatriol could inhibit, to a certain extent, Na^+^, K^+^- ATPase activity [79]. These results suggest the potential of panaxatriol’s anti-aging activity, although it was not as active as ABT-263 as a senolytic. In our cellular experiments, panaxatriol’s mechanism of action may be multi-targeted, and it still warrants further experiments to explore its anti-aging activity.

Preliminary experimental screening using the *C. elegans* models has been initiated. Although few results have been obtained so far, analysis of the results of the *C. elegans* lifespan experiments revealed an anti-aging effect of voclosporin, which has never before been tested for any anti-aging activity, demonstrating the value of the predicted results of this study in terms of drug repurposing. Of course, these are only the most preliminary results, and the mechanism through which voclospoin prolongs the lifespan of *C. elegans* needs to be further explored in subsequent experiments.

At the same time, voclosporin, which showed no senolytic activity in cellular experiments, could significantly extend the lifespan of *C. elegans*. On one hand, voclosporin, a calmodulin neural phosphatase inhibitor, synergistically inhibits T-cell-mediated immune responses, reducing the release of inflammatory mediators and the onset of inflammatory responses [80]. The anti-inflammatory effects of voclosporin may down-regulate the SASP, thereby mitigating the effects of aging. On the other hand, senolytics are usually cell type-specific and different induced senescence conditions lead to cellular senescence with different sensitivities to the drugs. The senolytic activity in our training dataset was determined in different cell lines; it would be one-sided to characterize the activity of compounds only on a single cell line model of induced senescence. A comprehensive and reliable senolytic activity assay relies on multiple cell lines, as well as on models using different senescence conditions, which is a shortcoming of this study and a much-needed direction for improvement in future screening efforts. The advantage of the *C. elegans* model is its ability to holistically assess the anti-aging activity of a drug at the individual level. However, for senolytics, simple lifespan measurements do not directly indicate whether drugs achieve their anti-aging effects through clearing specific senescent cells or identifying which types are targeted. Further experimentation is essential to clarify these aspects, and it will be central to our subsequent research initiatives.

## 4. Methods

### 4.1. Construction of Senolytic and Screening Datasets for Machine Learning

We compiled a dataset of characterized senolytics by querying the literature published before February 2024, focusing exclusively on small molecules with well-defined chemical structures to obtain their SMILES representations. These senolytics were validated across various cell lines, and despite the inherent heterogeneity, we included them to assemble a comprehensive activity dataset due to the limited number of known senolytics. Additionally, we curated a collection of compounds previously identified as non-senolytic, which included those screened experimentally and confirmed to be inactive in at least one cell line, as well as compounds effectively used as inactive datasets in virtual screening studies [27]. Given the rarity of senolytics, we assumed that the negative dataset contained no or very few potential senolytics. To further refine the negative dataset, we calculated the ECFP fingerprints (2048 bits, radius 2) of the compounds using the RDKit package (v2023.9.4, https://github.com/rdkit/rdkit/releases, accessed on 21 April 2025) and excluded those with a structural similarity greater than 0.9 to any compound in the positive dataset, based on the structural similarity comparison. The filtered dataset was then used as the negative dataset for training purposes (Appendix A).

The Drugbank dataset was downloaded from the Drugbank database (https://go.drugbank.com/releases, accessed on 18 June 2023, version 5.1.10) [62], and for small-molecule compounds without structural information, we used pubchempy (v1.0.4) to obtain structural information by drug name and CAS number. The TCMbank dataset was downloaded from the TCMbank database (https://tcmbank.cn/, accessed on 3 May 2024) [63]. We selected small-molecule compounds with SMILES information as our screening dataset. In the end, the Drugbank dataset contained 12,056 compounds and the TCMbank dataset contained 29,902 compounds.

### 4.2. Selection of Machine Learning Models and Molecular Features

To construct senolytic predictors with good performance, we evaluated popular models such as support vector machines and random forest in scikit-learn (v1.3.2), the extreme gradient boosting tree model in xgboost (v2.0.3), and pytorch (v2.1.0)-based constructs of multilayer perceptron machines and convolutional neural network models. Additionally, we tested deep learning models with more complex structures, such as the message-passing graphical neural network (MPNN) developed by Wang et al. [26] and the graphical neural network (GNN) developed by Tsubaki et al. [81]. However, their performance was not satisfactory, and thus we did not adopt them in our subsequent studies (Appendix A).

For molecular characterization, it has been previously shown that compound activity prediction models based on FP2 or ECFP molecular fingerprint features have good prediction performance [57]. Therefore, we used pybel, a Python wrapper for openbabel (v3.1.1), to compute and generate 1024-bit FP2 molecular fingerprint features from the SMILES strings of the compounds. The ECFP molecular fingerprint features were computed from the Rdkit package with a length of 2048 bits, and the radii were set as 2, 3, and 4, with the best model performance achieved at a radius of 2, which was finally adopted. The chemical molecular descriptors were adopted from previous senolytic prediction models [26,27]. Therefore, we also tried to construct a feature vector consisting of 200 molecular feature descriptors as the feature input of the molecules, which were also obtained by calculating the SMILES strings of the compounds using the Rdkit package. In recent years, molecular feature embeddings obtained from neural network language models have given good results in many downstream tasks, so we used the pre-trained neural network model MoLFormer, constructed by the IBM team, to generate molecular feature vectors [58].

To evaluate the model performance, the optimal hyperparameters were determined via a 5-fold cross-validation grid search based on stratified sampling, and the evaluation metrics mainly focused on the AUC and F1 score. For the random forest model, we adjusted the number of decision trees (n_estimators), and the other parameters were set to default values. For the support vector machine model, the kernel function (kernel) and the penalty coefficient (C) were adjusted. For the XGBoost model, the learning_rate, the number of decision trees (n_estimators), and the maximum depth (max_depth) were adjusted. For the deep learning models, MLP and CNN, the number of network layers, the number of hidden layer neurons, and the learning rate were adjusted. For the CNN model, we additionally adjusted the size of the convolutional kernel; for the model with only one convolutional layer, we adjusted the size of the convolutional kernel to 3, 5, 7, and 9; and for the model with multiple convolutional kernels, we adjusted the size of the convolutional kernel of the first convolutional layer to 3, 5, 7, and 9 (Appendix A). The performance impact of convolutional layers was evaluated using CNN models based on MoLFormer molecular embeddings. During model optimization, we employed the Adam optimizer with a learning rate of 0.0001, with a dropout of 0.5 and training for 3000 epochs with early stopping (patience = 100). A comparative analysis revealed that increasing the convolutional layer count consistently degraded performance, with optimal results achieved without convolutional layers (Appendix A).

For the dataset imbalance problem, we tried to use the data preprocessing technique SMOTE, which was applied only to the training set after data segmentation. In all five machine learning models that we used, SMOTE did not significantly improve the performance of the models and generally decreased the precision. It was prudent not to have too many false-positives in the predicted results, so our senolytic predictor did not use SMOTE.

### 4.3. Optimization of the SVM and MLP Models to Build Senolytic Predictor

Our senolytic predictor consisted of two models, SVM and MLP. For the SVM model, the optimal combination of hyperparameters was finally set as {‘C’: 20, ‘kernel’: ‘rbf’, ‘class_weight’: ‘balanced’}, and the other hyperparameters were set to default values. For the MLP model, the Adam optimizer and the cross-entropy loss function were used. We adjusted the number of hidden layers and neurons of the model and found that using too many hidden layers and neurons did not improve the performance of the model, so the final MLP model used had only one hidden layer (Appendix A). We finally adjusted the learning rate to 0.0001, set the epoch to 3000, and prevented the model from overfitting by introducing an early stopping mechanism that stopped training early if the model’s loss did not decrease in the next 100 epochs. Then, to further improve the reliability and generalization ability of the model, the ensemble strategy was used to generate 10 sub-models, and the prediction average of these sub-models was taken as the final prediction result.

### 4.4. Positive Sample Oversampling with the MolFormer Model

The MolFormer model generated 768-dimensional molecular embeddings as float vectors. Due to its inherent property of random initialization, the MolFormer model produces different molecular embeddings for the same molecule when processed in different batches. In addition, different random seeds can produce different molecular representations of the same molecule. Although these representations retain an intrinsic similarity, they are treated as distinct molecules by downstream machine learning models, which can affect model performance and prediction accuracy. Leveraging this characteristic, and considering the scarcity of positive data in our training set, we applied oversampling to the positive samples using the MolFormer model. The process was straightforward, described as follows: for the 111 positive samples in the training set, molecular embeddings were generated 5 times without setting a fixed random seed, resulting in a positive dataset containing 555 samples. This oversampling approach aimed to allow the classification model to better learn the features of positive compounds and mitigate the effects of batch variability. However, it was critical to carefully control the degree of oversampling to avoid overfitting. The five-fold oversampling achieved an appropriate balance. It had an excellent test set performance while maintaining a stable training set AUC of 0.99, despite a modest precision decrease as oversampling increased (Appendix A).

### 4.5. Evaluation Metrics for Model Performance

All models were evaluated using the following classification performance metrics [82]:Accuracy = (TP + TN)/(TP + FN + FP + TN)Precision = TP/(TP + FP)Recall = TP/(TP + FN)F1 score = 2 × (Precisions × Recall)/(Precision + Recall)
where TP, TN, FP, and FN are the number of true-positives, true-negatives, false-positives, and false-negatives in the model’s predictions, respectively. We focused mainly on the F1 score [83], which is the reconciled average of precision and recall, and is a measure of a classification model’s ability to predict positive classes in the case of positive–negative sample imbalance. The other metric was AUC [84], a metric widely used for classification model evaluation that reflects the model’s ability to classify at different thresholds, and is not affected by the proportion of positive and negative samples.

For plotting the precision–recall curve, we refer to the method of Wong et al. [26], and used the precision_recall_curve module in scikit-learn to calculate the value of auPRC [85], and through the bootstrap method, we performed 100 bootstraps to plot the precision–recall curve and calculate the 95% confidence interval.

### 4.6. t-SNE and Molecular Similarity Comparison

For the t-SNE analyses shown in Figure 4c and Figure 5c, the t-SNE module in scikit-learn was used in conjunction with the MolFormer representations of the compounds for visualization. The metric distance was used as the cosine distance and the perplexity parameter was set to 10 to obtain plots with appropriately spaced data points. For the t-SNE analysis shown in Appendix A, the ECFP molecular fingerprint (2048 bits, radius 2) representation of the compounds and the Jaccard distance were used to metricize the differences in the molecular structures. The Jaccard distance is also known as the Tanimoto distance and is defined as the 1-Tanimoto similarity. The comparison of molecular fingerprint similarity in this study was also measured using ECFP molecular fingerprint (2048 bits, radius 2) and Tanimoto similarity (Figure 4d and Figure 5d).

### 4.7. Enrichment Analysis of the TCMbank Prediction Results

We mapped the 714 potential senolytics obtained from the joint prediction of MLP and SVM to the corresponding medicinal plant sources. Enrichment analysis was performed using the enricher function in the clusterProfiler (v4.10.0) package for R (v4.4.3), sorted by P_adj. For KEGG enrichment analysis of target genes [68], online enrichment analysis was also performed using the clusterProfiler package. All of these enrichment results were visualized using ggplot2 (v3.4.4) to plot the top 20 results.

### 4.8. Analysis of the Applicability Domain of Models

The predictive ability of QSAR models is limited by the chemical space occupied by the training set [86]. If the chemical structures of the query compounds are too different from the compounds in the training set, the prediction results of the model will inevitably be highly biased. Therefore, any QASR model should carefully consider the range of the applicable domain (AD). In this study, we used a Euclidean distance-based approach to calculate the AD. The applicability domain threshold (ADT) was obtained by calculating the Euclidean distance distribution matrix of the training set [87]. Then, the distance between the query compound and its nearest neighbor in the training set was taken as the applicable distance, and if the distance was less than the ADT, then the compound was considered to be within the AD and its prediction was relatively reliable, and vice versa, so the prediction was considered to be unreliable. The formula used for calculating the applicable domain distance is as follows:ADT = D + Zσ
where Z is the empirical similarity threshold parameter (default value is 0.5), while D and σ are the mean and standard deviation of all Euclidean distances in the multidimensional descriptor space for all compounds in the training set, respectively. We use the MolFormer embedding as the molecular representation for calculating the AD, which was consistent with our senolytic predictor. Moreover, the results of the AD calculations are shown in Appendix A.

### 4.9. Cell Line and Senescence Induction

The IMR-90 human female fetal lung fibroblast cell line was obtained from the National Collection of Authenticated Cell Cultures (Shanghai, China; Catalog No. GNHu32). Cells were routinely cultured in complete Dulbecco’s Modified Eagle Medium (DMEM; Gibco, Thermo Fisher Scientific, Waltham, MA, USA) supplemented with 10% fetal bovine serum and 1% penicillin–streptomycin solution (Thermo Fisher, Waltham, MA, USA) at 37 °C in a 5% CO_2_ humidified incubator, with fresh medium replaced every 48 h. For senescence induction, etoposide (MedChemExpress, Monmouth Junction, NJ, USA) was first dissolved in DMSO to prepare a 50 mM stock solution, which was then diluted and aliquoted into a 10 mM working concentrate. Following passaging, when cells reached 30–50% confluency, the 10 mM etoposide solution was added to the culture medium at a dilution of 1:200, achieving a final working concentration of 50 μM. Cells were exposed to etoposide for 48 h to induce senescence and subsequently allowed them to recover for 72 h in fresh complete DMEM. These cells were then considered to be senescence-induced. In parallel, control cells were treated with 0.5% DMSO for 48 h, following the same protocol, to serve as the normally proliferating control group.

### 4.10. SA-β-Gal Staining

Senescence-associated β-galactosidase (SA-β-gal) activity was assessed using a β-gal staining kit (Solarbio, Beijing, China). Cells were seeded in 6-well plates and washed once with PBS. Subsequently, 1 mL of β-Gal fixation solution was added per well, and the cells were fixed at room temperature for 15 min. After fixation, cells were washed three times with PBS, and 1 mL of staining solution was added to each well. The plates were then incubated overnight (approximately 12 h) in a 37 °C humidified incubator. Following incubation, the staining solution was removed and 2 mL of PBS was added per well. The stained cells were immediately observed under a light microscope (Leica, Wetzlar, Germany) at 5× magnification, and representative images were captured.

### 4.11. Cell Viability Assay

For the cell viability assay, cells were seeded in 10 cm culture dishes and allowed to grow until reaching 80–90% confluency. Cells were then detached using trypsin, resuspended in fresh medium, and counted using a hemocytometer. The cell suspension was diluted to approximately 1 × 10^5^ cells/mL, and 100 μL of the suspension was dispensed into each well of a 96-well plate. After allowing 24 h for cell attachment and growth, the test compounds were added. All compounds (obtained from MedChemExpress, Monmouth Junction, NJ, USA) were prepared as 10 mM stock solutions in DMSO and diluted in culture medium to the required working concentrations at the time of the assay. ABT-263, a known senolytic agent, was used as a positive control, and it was also prepared as a 10 mM stock solution and further diluted 1:1000 (to yield 10 μL working volume) for the experiments. The negative control group received 0.1% DMSO. Each treatment group was performed in triplicate. Following the addition of compounds, cells were incubated for 72 h in the cell culture incubator. Cell viability was subsequently determined using the CCK8 assay (Servicebio, Beijing, China). CCK8 was diluted 1:10 in the culture medium, and 100 μL of the diluted solution was added to each well. After a 40-min incubation in the incubator, absorbance was measured using an M200PRO multifunctional microplate reader (Tecan, Mannedorf, Switzerland) at 450 nm, with 620 nm serving as the reference wavelength. The cell viability was calculated by subtracting the background signal measured at 620 nm.

### 4.12. C. elegans Strains and Maintenance

The wild-type strain, N2, used in this study was obtained from the Caenorhabditis Genetics Center (CGC, University of Minnesota, Minneapolis, MN, USA). All strains were cultivated on nematode growth medium (NGM) plates seeded with live *E. coli* OP50 as food and incubated at 20 °C. The bleach (0.5 M NaOH, 5% NaClO) was used to separate the fertilized egg from the mother and to keep the offspring in the same lifetime. *E. coli* OP50 was cultured in LB liquid medium, shaken at 37 °C for 24 h, and concentrated by centrifugation for 15×.

### 4.13. C. elegans Lifespan Measurement

We refer to the approach of Clay et al. [88]. In a 96-well plate, 200 μL of each component was added to each well, including approximately 10 to 20 L4 stage nematodes transferred from NGM medium, S medium, *E. coli* OP50, fluorouridine (FUdR, Tansoole, Shanghai, China), and the test compound. All compounds, except metformin (Leyan, Shanghai, China), are soluble in DMSO. The concentrations of fluorouridine and metformin were set at 0.1 mM, while the concentrations of all DrugBank compounds (MedChemExpress, Shanghai, China) were 0.03 mM, with DMSO having a concentration of 3 ‰. Each control group utilized three wells to ensure the reliability and reproducibility of the results through triplicate measurements.

## 5. Conclusions

In summary, the presented senolytic predictor performed well, with an AUC score of over 0.99, and significantly outperformed previous models. This was achieved by using an expanded dataset and more extensive molecular characterization evaluations. Furthermore, the model predicted hundreds of compounds with potential senolytic activity and identified 81 medicinal herbs with possible senolytic properties. Finally, the validity of the model was confirmed in subsequent in vitro cellular and nematode lifespan experiments. The results showed that panaxatriol exhibited senolytic activity on the etoposide-induced senescence of the IMR-90 cell line. Voclosporin significantly extended the lifespan of *C. elegans*. This study not only provides an efficient framework for discovering novel senolytic agents, but also highlights the predicted novel senolytic compounds and herbs as valuable starting points for future senolytic drug development.

Senolytics that target and eliminate senescent cells act through diverse mechanisms that are not yet fully understood. This underscores the value of target-agnostic machine learning strategies in advancing drug discovery [89]. In the future, further experimental validation and model optimization will drive more accurate predictions, thereby facilitating the early discovery of senolytics and reducing experimental costs. Moreover, the improved interpretability of predictions may be achieved by incorporating insights into the biological targets and mechanisms underlying senolytic activity.

## Figures and Tables

**Figure 1 molecules-30-02653-f001:**
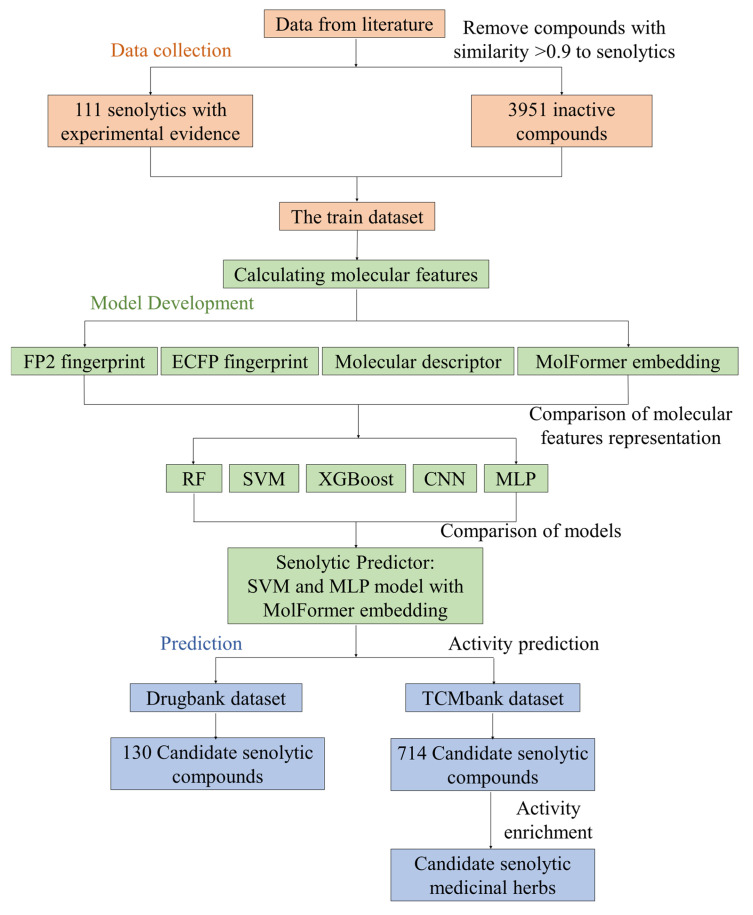
The workflow used for developing the senolytics predictor. Active and inactive compound data were obtained from the literature to assemble the training set. For the activity dataset, small-molecule compounds that were validated on at least one cell line to exhibit senolytic activity at concentrations no greater than 10 μmol were selected. For the inactive dataset, small-molecule compounds used as inactive compounds in previous screening efforts, or those that did not exhibit senolytic activity, were selected. Finally, our senolytic predictor was composed of SVM and MLP models for virtual screening. Potential senolytics were obtained by predicting the DrugBank and TCMbank datasets, and then we identified the medicinal herbs that may have senolytic activity by enrichment analysis.

**Figure 2 molecules-30-02653-f002:**
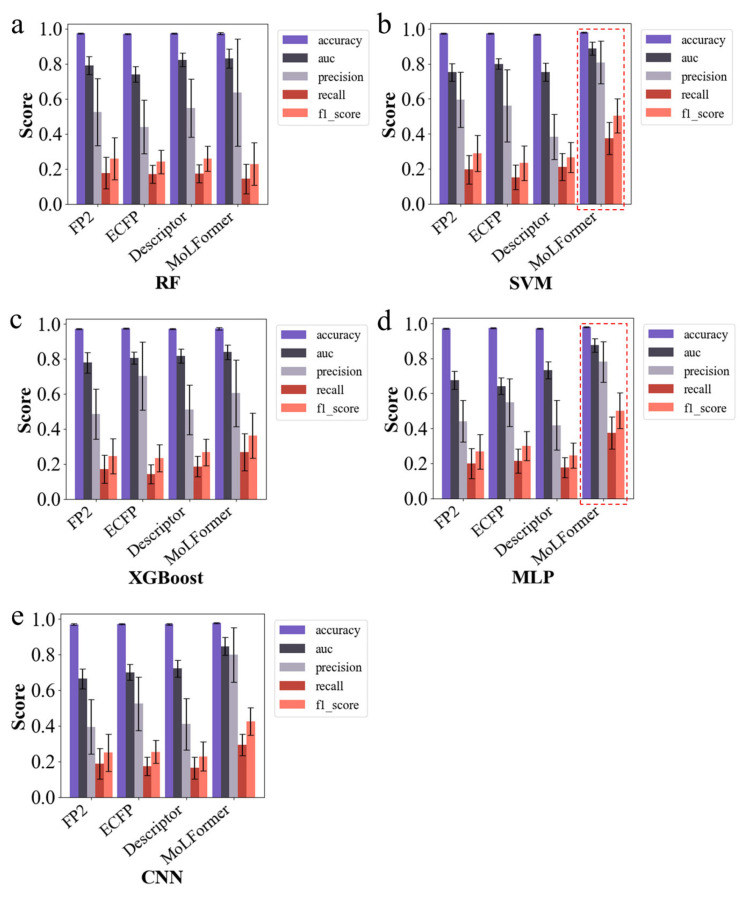
Selection of molecular features and machine learning models. (**a**–**e**) Show the performance of each classification model. These models include RF (**a**), SVM (**b**), XGBoost (**c**), MLP (**d**), and CNN (**e**), using FP2 molecular fingerprints, ECFP molecular fingerprints, molecular descriptors, and MolFormer embeddings as inputs to the models, respectively. The values of the evaluation metrics are the average of 20 iterations of repeated random-split cross-validation, and the evaluation metrics include accuracy, AUC, precision, recall, and F1 score.

**Figure 3 molecules-30-02653-f003:**
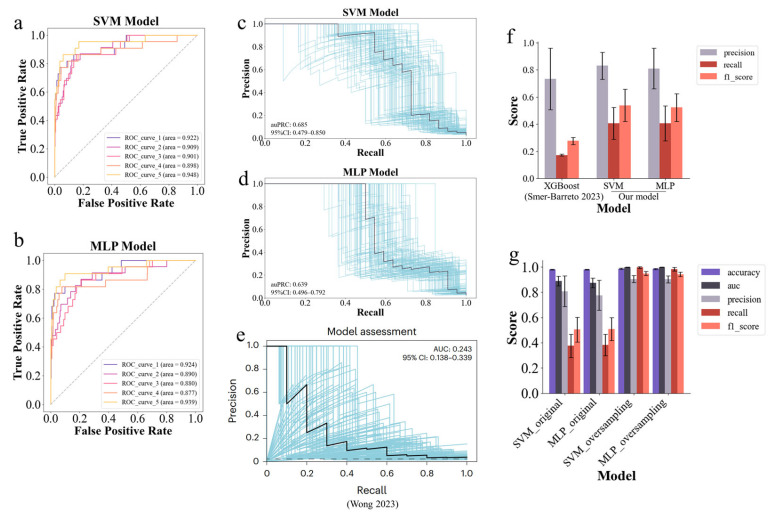
Construction of senolytic predictor. (**a**,**b**) AUCs for the SVM and MLP models, respectively. AUCs were obtained for each occasion via 5-fold cross-validation based on stratified sampling. (**c**,**d**) Precision–recall curves for the SVM and MLP models, respectively. Gray curves and 95% confidence intervals (CIs) indicate the changes generated via bootstrapping, as in (**e**). (**e**) Precision–recall curves for the MPNN senolytic activity prediction model of Wong et al. [26]. The figure is taken directly from that paper. (**f**) A performance comparison of our SVM and MLP models with the senolytic prediction model of Smer-Barreto et al. [27]. For the performance comparisons, we used the same methodology, i.e., 5-fold cross-validation, to evaluate the performance of the models. The model performance data of Smer-Barreto et al. [27]. were taken directly from the data presented in that paper. (**g**) Model performance using raw data or oversampling. The methods and metrics used to assess model performance in the figure are the same as in Figure 2.

**Figure 4 molecules-30-02653-f004:**
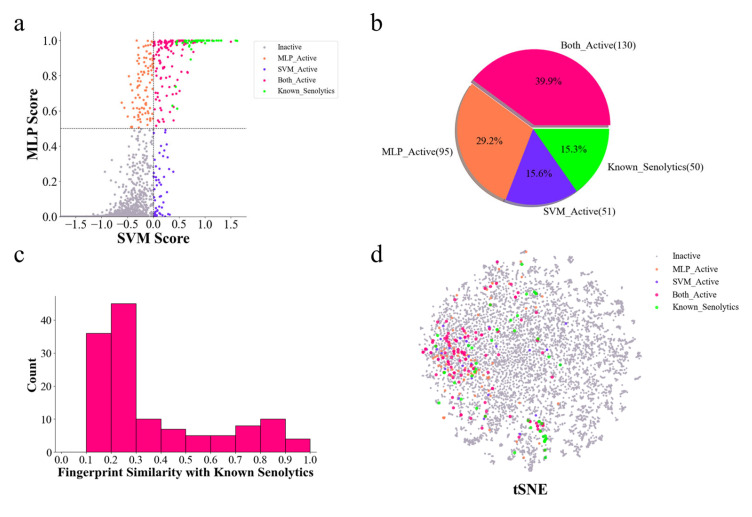
The discovery of new senolytics through DrugBank dataset prediction. (**a**) The MLP and SVM model predicted probabilities of senolytic activity for all compounds in the DrugBank dataset. Each dot indicates one compound. Red and gray dots indicate compounds jointly predicted as active and inactive senolytics via the MLP and SVM models, respectively. Orange dots indicate that only the MLP model predicts as senolytics. Blue dots indicate that only the SVM model predicted as senolytics. Green dots indicate that these compounds are known senolytics in the DrugBank dataset that are also in our training set. (**b**) A pie chart of the distribution of the number of compounds predicted as senolytics using the models and the known senolytics in the Drugbank dataset, corresponding to (**a**). (**c**) A histogram of the distribution of the 130 compounds predicted as senolytics by the two models together in (**b**) compared to the Tanimoto similarity of the ECFP fingerprints of the known senolytics in the training set. (**d**) t-SNE plot of the chemical spatial distribution of the Drugbank dataset. Each dot represents a compound, and the color of the dot corresponds to (**a**).

**Figure 5 molecules-30-02653-f005:**
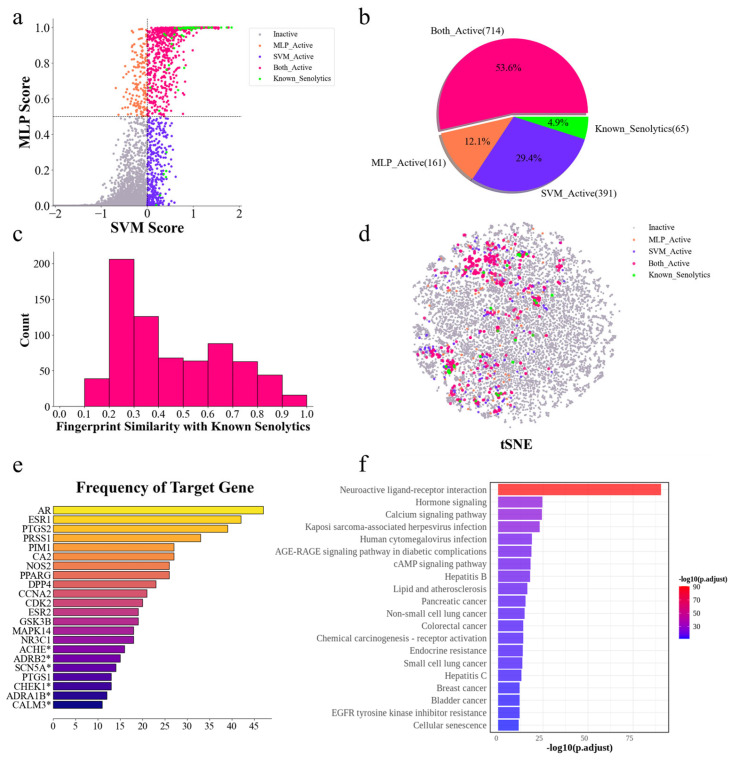
The identification of potential senolytic medicinal herbs through prediction on the TCMbank dataset. (**a)** The MLP and SVM models predicted probabilities of senolytic activity for all compounds in the TCMbank dataset. Each dot indicates one compound. Red and gray dots indicate compounds jointly predicted as active and inactive senolytics using the MLP and SVM models, respectively. Orange dots indicate that only the MLP model predicts as senolytics. Blue dots indicate that only the SVM model predicted as senolytics. Green dots indicate that these compounds are known senolytics in the TCMbank dataset, which are also in our training set. (**b**) A pie chart of the distribution of the number of compounds predicted as senolytics using the models and the known senolytics in the TCMbank dataset, corresponding to (**a**). (**c**) A Histogram of the distribution of the 714 compounds predicted as senolytics using the two models together in (**b**) compared to the Tanimoto similarity of the ECFP fingerprints of the known senolytics in the training set. (**d**) A t-SNE plot of the chemical spatial distribution of the TCMbank dataset. Each dot represents a compound, and the color of the dot corresponds to (**a**). (**e**) Statistics of the number of occurrences of target genes for the potential senolytics, showing target genes with more than 10 hits. * Indicates that these target genes are not targets of known senolytics. (**f**) KEGG enrichment analysis using all target genes of the potential senolytics.

**Figure 6 molecules-30-02653-f006:**
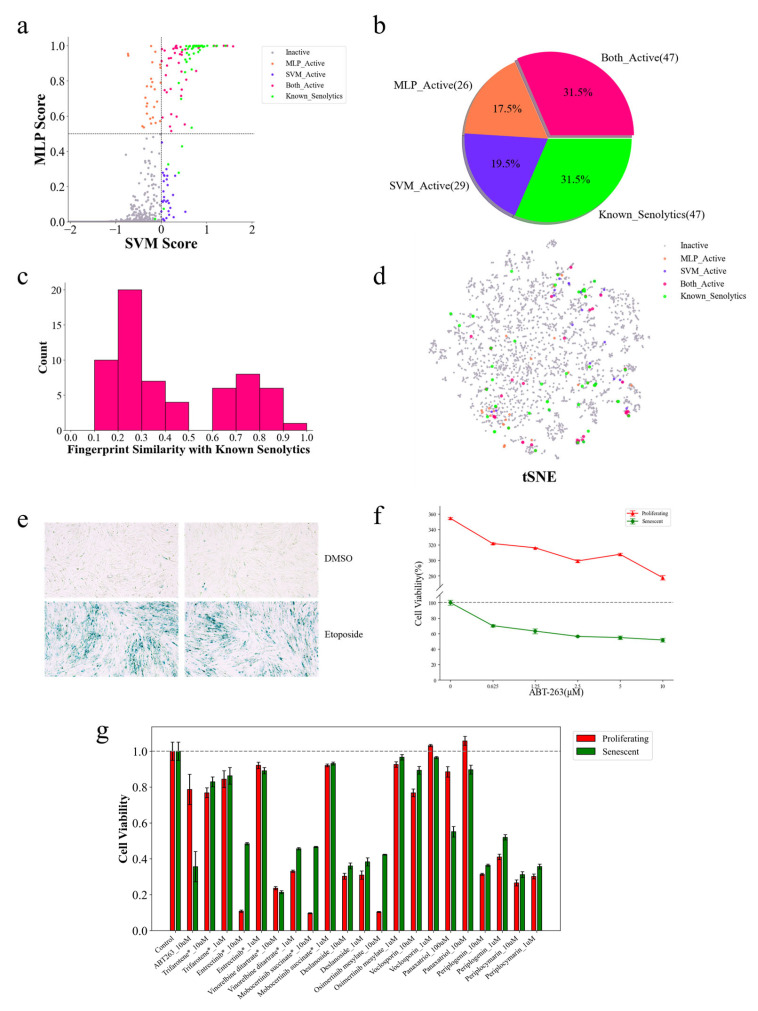
The discovery of candidate senolytics through the FDA_HY-L022 and TCMM_HY-L065 datasets prediction. (**a**) The probability of MLP and SVM models predicting senolytic activity for all compounds in the query dataset. (**b**) A pie chart of the distribution of the number of compounds predicted using the models to be senolytic drugs versus the known senolytic compounds in the query dataset, with the colors corresponding to (**a**). (**c**) A histogram of the distribution of the structural similarity of 62 compounds predicted using the two models together. The structural similarity was calculated against the known senolytic drugs in (**b**). (**d**) A t-SNE plot of the chemical spatial distribution of the query dataset. (**e**) SA-β-gal staining of proliferating and induced senescent cells. (**f**) Activity test of different concentrations of ABT-263. (**g**) Senolytic activity testing of candidate compounds. * Indicates that these compounds were predicted to be negative using the models and were tested as negative controls. ABYT-263 was used as a positive control and the remaining 6 compounds were candidates predicted to be active using the models.

**Figure 7 molecules-30-02653-f007:**
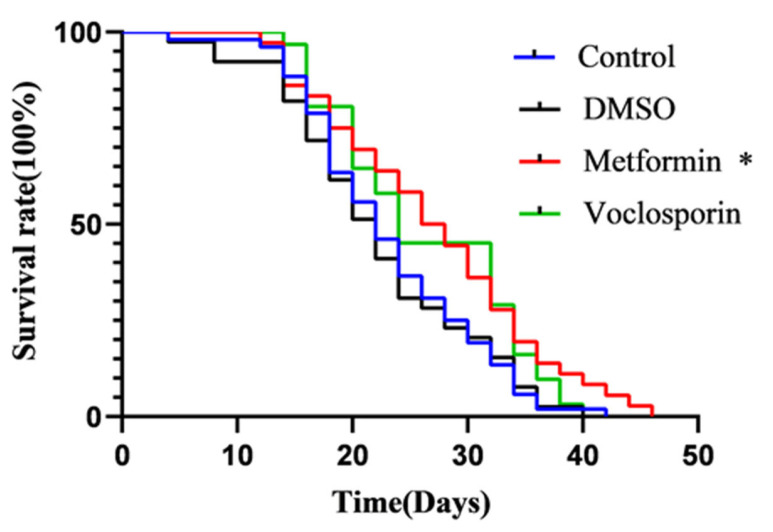
The survival curve of *C. elegans*. * Indicates that the survival curve is significantly different compared to that of the DMSO group (*p*-value < 0.05).

**Table 1 molecules-30-02653-t001:** The top 10 compounds in the DrugBank dataset predicted to have the highest score using both MLP and SVM models.

Name	Match_degree ^1^	Match_id ^2^	SVM Score	SVM Prediction	MLP Score	MLP Prediction	Target
Cymarin	0.793	Strophanthin K	1.501	1	0.993	1	/
Metildigoxin	0.889	Digoxin	1.140	1	0.999	1	/
Amrubicin	0.575	Idarubicin	1.057	1	1.000	1	TOP2A
Alisporivir *	0.783	Cyclosporin A	0.972	1	1.000	1	CAMLG
Reidispongiolide A *	0.196	Alvespimycin	0.929	1	0.991	1	ACTA1
NIM811 *	0.924	Cyclosporin A	0.865	1	1.000	1	/
Acetyldigitoxin	0.846	Digitoxin	0.816	1	0.996	1	ATP1A1
Aclarubicin	0.339	Idarubicin	0.804	1	0.990	1	TOP2A, TOP2B
Acetyldigoxin	0.846	Digoxin	0.767	1	0.986	1	/
Quisinostat	0.245	CUDC-907	0.748	1	1.000	1	/

Note: The top 10 list represents the compounds with the highest sum of scores modeled by SVM and MLP. * Indicates that the known targets of these compounds are different from known senolytics. ^1^ The similarity value of the ECFP fingerprint between the query compound and the match compound. ^2^ The compound in the active training set has the highest similarity with the query compound.

**Table 2 molecules-30-02653-t002:** The top 10 compounds in the TCMbank dataset predicted to have the highest score using both the MLP and SVM models.

Name	Match_degree ^1^	Match_id ^2^	SVM Score	SVM Prediction	MLP Score	MLP Prediction
Sesguoiaflavone	0.705	Ginkgetin	1.766	1	1.000	1
Bufotalidin	0.656	Strophanthidin	1.757	1	1.000	1
Helveticoside	0.789	Convallotoxin	1.553	1	0.999	1
cis-Miyabenol a	0.274	Procyanidin C1	1.541	1	1.000	1
Bipindoside	0.756	Ouabain	1.545	1	0.993	1
Cymarin	0.793	Strophanthin K	1.501	1	1.000	1
Malayoside	0.878	Peruvoside	1.493	1	1.000	1
Helveticosol	0.688	Digitoxin	1.485	1	0.995	1
Cymarol	0.750	Periplocin	1.472	1	0.997	1
Gnetuhainin m	0.333	Procyanidin C1	1.460	1	1.000	1

Note: The top 10 list represents the compounds with the highest sum of scores modeled via SVM and MLP. ^1^ The similarity value of the ECFP fingerprint between the query compound and the match compound. ^2^ The compound in the active training set has the highest similarity with the query compound.

**Table 3 molecules-30-02653-t003:** The top 20 potential herbs with senolytic activity obtained by enrichment analysis.

Herb_name	Family	Genus	Compound	P_adj ^1^
*Bufo bufo gargarizans, Bufo melanostictus*	Bufonidae	Bufo	23	2.92 × 10^−19^
*Erysimum cheiranthoides*	Brassicaceae	Erysimum	9	8.75 × 10^−11^
*Strophanthus divaricatus*	Apocynaceae	Strophanthus	10	2.10 × 10^−10^
*Thevetia neriifolia*	Apocynaceae	Thevetia	10	3.20 × 10^−8^
*Corchorus capsularis*	Malvaceae	Corchorus	6	3.20 × 10^−8^
*Morus alba L.*	Moraceae	Morus	22	3.20 × 10^−8^
*Strophanthus kombe*	Apocynaceae	Strophanthus	7	3.83 × 10^−8^
*Tabernaemontana corymbosa*	Apocynaceae	Tabernaemontana	7	1.09 × 10^−7^
*Corchorus olitorius*	Malvaceae	Corchorus	5	2.84 × 10^−7^
*Cerbera manghas*	Apocynaceae	Cerbera	6	9.56 × 10^−7^
*Cerbera odollam*	Apocynaceae	Cerbera	5	2.12 × 10^−6^
*Erysimum diffusum*	Brassicaceae	Erysimum	4	2.55 × 10^−6^
*Rosa chinensis*	Rosaceae	Rosa	16	2.60 × 10^−6^
*Antiaris toxicaria*	Moraceae	Antiaris	5	7.31 × 10^−6^
*Adonis mongolica*	Ranunculaceae	Adonis	4	1.01 × 10^−5^
*Bupleurum smithii*	Apiaceae	Bupleurum	5	1.96 × 10^−5^
*Nerium indicum*	Apocynaceae	Nerium	11	2.52 × 10^−5^
*Gnetum gnemon*	Gnetaceae	Gnetum	5	4.31 × 10^−5^
*Adonis vernalis*	Ranunculaceae	Adonis	4	5.44 × 10^−5^
*Bupleurum yinchowense*	Apiaceae	Bupleurum	3	0.000102922

Note: Ranking based on P_adj values. ^1^ The P_adj value is the adjusted *p*-value used to control the false discovery rate in multiple tests. A P_adj value < 0.05 indicates statistical significance.

**Table 4 molecules-30-02653-t004:** The *C. elegans* lifespan assay of 5 compounds from DrugBank.

NAME	Structure	SVM Score	SVM Prediction	MLP Score	MLP Prediction	Life Expectancy (Day)	Life Extension Rate(%)	*p* Value ^1^
Voclosporin	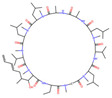	0.746	1	0.999	1	26.39 ± 1.53	19.1	0.060
Simeprevir	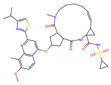	−0.654	0	0.002	0	19.52 ± 0.86	−11.9	0.054
Belinostat	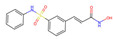	−0.544	0	0.024	0	22.77 ± 0.62	2.8	0.760
Paritaprevir	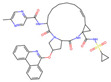	−0.743	0	0.0	0	22.53 ± 1.96	1.7	0.870
Tenapanor	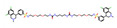	−0.074	0	0.098	0	23.37 ± 1.58	5.5	0.338
Control	/	/	/	/	/	23.15 ± 0.15	/	
DMSO	/	/	/	/	/	22.15 ± 0.17	/	
Metformin	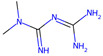	/	/	/	/	25.78 ± 2.40	11.3	0.039

Note: ^1^ The *p*-values indicate the significance of the survival curve, calculated using the Wilcoxon test.

## Data Availability

The source code and data accompanying this study are publicly available at https://gitee.com/lijinjunkust/senolytic-predictor, accessed on 21 April 2025.

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
