# Peer review of "Development and Application of a Senolytic Predictor for Discovery of Novel Senolytic Compounds and Herbs"

_molecules, 2025, doi:10.3390/molecules30122653_

Round 1
Reviewer 1 Report (Previous Reviewer 3)
Comments and Suggestions for Authors
I appreciate the authors’ substantial improvements, particularly the addition of experimental validation, which was previously lacking and formed the basis of my initial recommendation for rejection. The new data significantly strengthen the study’s conclusions and support the utility of the proposed senolytic predictor. I now find the manuscript suitable for publication in Molecules.
Author Response
Reviewer 1 comments and response
Comments 1: I appreciate the authors’ substantial improvements, particularly the addition of experimental validation, which was previously lacking and formed the basis of my initial recommendation for rejection. The new data significantly strengthen the study’s conclusions and support the utility of the proposed senolytic predictor. I now find the manuscript suitable for publication in Molecules.
Response 1: Thank you very much for your thoughtful and constructive feedback on our manuscript. We are grateful for the time and effort you have put into reviewing our work. Your substantial suggestions have contributed to the improvement of the article and we are also encouraged by your recommendation for publication.

Reviewer 2 Report (New Reviewer)
Comments and Suggestions for Authors
The manuscript titled ‘’ Development and Application of a Senolytic Predictor for Discovery of Novel Senolytic Compounds and Herbs ‘’ present an interesting work. The authors attempted to develop a new predictive model that is better than existing models in terms of performance. But there are a few points that should be considered and, if deemed appropriate, included in the manuscript by the authors to improve the quality of the manuscript.
In the introduction, the authors state '' Notably, the precision of our MLP and SVM models reached 0.904 and 0.903, respectively. Using this predictor, we screened the DrugBank dataset and predicted 130 predicted senolytic compounds as drug repurposing candidates. Please do not present the results in the introduction.
Why did you use compounds with missing experimental data as inactive datasets ? Will this not affect the quality of the models developed ?
Please express the meaning of all abbreviations you have cited in your manuscript.
Please add references in the paragraph defining F1 score and AUC.
Please avoid presenting the results in the methods section, as you did in part ''4.8. Analysis of Applicability Domain of Models''
Provide a comprehensive conclusion, separate from the discussion section, highlighting the main findings
Author Response
Reviewer 2 comments and response
Comments 1: The manuscript titled “ Development and Application of a Senolytic Predictor for Discovery of Novel Senolytic Compounds and Herbs ” present an interesting work. The authors attempted to develop a new predictive model that is better than existing models in terms of performance. But there are a few points that should be considered and, if deemed appropriate, included in the manuscript by the authors to improve the quality of the manuscript.
Response 1: Thank you for your positive feedback and recognizing the value of our work. Your comments and suggestions are helpful in improving our manuscript. We have carefully revised the manuscript based on the comments and the detailed responses are as follows.
Comments 2: In the introduction, the authors state '' Notably, the precision of our MLP and SVM models reached 0.904 and 0.903, respectively. Using this predictor, we screened the DrugBank dataset and predicted 130 predicted senolytic compounds as drug repurposing candidates. Please do not present the results in the introduction.
Response 2: Thank you for your advice. We adjusted this section. By removing the description of the results, this part was brought into line with the ‘introduction’ specification.
Comments 3: Why did you use compounds with missing experimental data as inactive datasets ? Will this not affect the quality of the models developed ?
Response 3: Thank you for your insightful comment. We sincerely apologize for the potential misunderstanding caused by our description in the manuscript. To clarify, the compounds included in our inactive dataset were carefully curated to ensure their reliability. This collection consisted of compounds previously identified as non-senolytic, which were either screened experimentally and confirmed as inactive in at least one cell line or used effectively in virtual screening studies as inactive datasets by Smer-Barreto et al.[1] Additionally, to further ensure dataset quality, we excluded any compound from this negative set exhibiting >90% structural similarity (based on molecular fingerprints) to known senolytics. This minimized the risk of misclassifying latent positives. We have modified the presentation in the manuscript to minimize misunderstandings (lines 121 and 590).
[1] Smer-Barreto V, Quintanilla A, Elliott RJR, Dawson JC, Sun J, Campa VM, Lorente-Macías Á, Unciti- Broceta A, Carragher NO, Acosta JC, Oyarzún DA. Discovery of senolytics using machine learning. Nat Commun. 2023 Jun 10;14(1):3445.
Comments 4: Please express the meaning of all abbreviations you have cited in your manuscript.
Response 4: Thank you for your careful suggestions. As you suggested, we have examined the manuscript and added the full names corresponding to the abbreviations in the following sections: AUC (Area Under Curve) (line 25); auPRC (Area Under the Precision-Recall Curve) (line 95); FP2 (Open Babel FP2 fingerprints) (line 150); ECFP (Extended Connectivity Fingerprints) (line 150); SMILES (Simplified Molecular Input Line Entry System) (line 155); KEGG (Kyoto Encyclopedia of Genes and Genomes) (line 356).
Comments 5: Please add references in the paragraph defining F1 score and AUC.
Response 5: Thank you for your careful suggestions. We have added references to F1 score, AUC and auPRC in the appropriate places (lines 96, 97, 162).
Comments 6: Please avoid presenting the results in the methods section, as you did in part ''4.8. Analysis of Applicability Domain of Models''.
Response 6: Thank you for your advice. As you suggested, we have deleted the result section in part ''4.8. Analysis of Applicability Domain of Models'' and instead described them in sections 2.4 and 2.5 of the manuscript results section (lines 229-231, 277-279).
Comments 7: Provide a comprehensive conclusion, separate from the discussion section, highlighting the main findings.
Response 7: Thank you for your constructive suggestions. We have added a separate conclusion section following the discussion and highlighting the main findings of the manuscript (lines 567-579).

Reviewer 3 Report (New Reviewer)
Comments and Suggestions for Authors
The manuscript presents a valuable approach for discovering potential senolytics using machine learning and phenotypic data. However, several points warrant further consideration before the manuscript can be considered for publication:
- The authors mention dataset heterogeneity—could they specify the types (e.g., senescence inducers, assay systems) and discuss how these may differentially impact prediction across compound classes?
- The dual-model prediction criterion may reduce false positives but potentially lowers recall. Could the authors comment on whether this strategy misses true positives that are confidently predicted by only one model?
- Voclosporin extended lifespan in C. elegans but lacked senolytic activity in IMR-90 cells. Could alternative anti-aging mechanisms be discussed, beyond senescent cell clearance?
- The manuscript notes limited model interpretability. Have the authors considered post-hoc methods (e.g., SHAP) to identify key molecular features contributing to predictions?
- The MoLFormer-based oversampling improved performance but may risk overfitting. How was this assessed or controlled beyond sample ratio adjustment?
Author Response
Reviewer 3 comments and response
Comments 1: The authors mention dataset heterogeneity—could they specify the types (e.g., senescence inducers, assay systems) and discuss how these may differentially impact prediction across compound classes?
Response 1: Thank you very much for your valuable feedback. We have added your suggestion to the discussion section of the manuscript (lines 458-467): In previous senolytic drug discovery efforts, researchers have screened for senolytics using different senescent cell models, including different cell types such as IMR-90, HUVEC, and A549, among others, as well as different senescence strategies, such as replicative senescence and chemotherapeutic drug (e.g., adriamycin and etoposide) induced cell senescence. The senolytics identified in these screening efforts also have diverse targets and mechanisms, including BCL2 family protein inhibitors, BET family protein inhibitors, and P53 stabilizers, among others. Unfortunately, due to the small number of senolytics discovered so far, these heterogeneities are difficult to discuss, which likewise implies the value of target-agnostic machine learning strategies. Therefore, a phenotype-based strategy was used in this study to construct the prediction model.
Comments 2: The dual-model prediction criterion may reduce false positives but potentially lowers recall. Could the authors comment on whether this strategy misses true positives that are confidently predicted by only one model?
Response 2: Thank you for your constructive feedback. This strategy does miss true positives that are confidently predicted by only one model. But we chose this strategy for reasons of Cost-Benefit Justification. And we believe that it is worthwhile to make the model have fewer false predictions at the cost of reducing the recall by a certain amount. We have added this section to the discussion section of the manuscript. (lines 517-520)
Comments 3: Voclosporin extended lifespan in C. elegans but lacked senolytic activity in IMR-90 cells. Could alternative anti-aging mechanisms be discussed, beyond senescent cell clearance?
Response 3: Thank you for your advice. As you suggested, we have added a discussion of possible anti-aging mechanisms of voclosporin. Since there are no direct reports of anti-aging activity of voclosporin, we have only briefly discussed it. (lines 548-553)
Comments 4: The manuscript notes limited model interpretability. Have the authors considered post-hoc methods (e.g., SHAP) to identify key molecular features contributing to predictions?
Response 4: We appreciate your insightful suggestion regarding post-hoc interpretability methods such as SHAP. While we recognize the value of these techniques for traditional feature-based models, our approach employs deep learning-generated molecular embeddings as input features. These embeddings are 768-dimensional abstract representations learned by MolFormer, which lack direct correspondence to chemically meaningful substructures (e.g., functional groups or bond types). Interpretability problems are common to such models, and applying SHAP to such embeddings would yield contributions mapped to abstract dimensions, which cannot be translated into actionable chemical insights.
Comments 5: The MoLFormer-based oversampling improved performance but may risk overfitting. How was this assessed or controlled beyond sample ratio adjustment?
Response 5: Thank you very much for your valuable comments. We have added a more explicit description in Section 5.4 and in Supplementary Figure 7. Specifically, by comparing the performance of the model on the test set at different oversampling multiples, we find that the performance of the model with 5x oversampling slows down compared to 4x oversampling, implying that higher oversampling is less meaningful (Supplementary Fig. 7a,b). On the other hand, we assessed the overfitting risk of the model by predicting the training set. The training set features used for testing are generated by the MolFormer model under a new random seed, different from the feature representation used for training. As shown in Supplementary Fig. 7c, d, as expected, the model's recall improves and precision decreases as the oversampling multiplicity increases, but the decrease is not particularly large, and in addition, the model's AUC is able to stabilize at 0.99. Therefore, we believe that this is acceptable, and that the 5-fold oversampling is in an appropriate balance to be adopted.

Reviewer 4 Report (New Reviewer)
Comments and Suggestions for Authors
The manuscript presents a comprehensive critical analysis of the application of a predictor for the discovery of new compounds and secondary metabolites for novel senolytic agents. The study covers everything from selection to the effect of senescence on the IMR-90 cell model, allowing for confirmation of the prediction model and virtual screening. However, attention to the following points is recommended:
L-23 Eliminate personal language such as "we" to present an impersonal presentation.
L-112 Include the working hypothesis. Although the objective of the work is presented based on the review in the introduction, guidelines could be provided for generating the working hypothesis, including the characteristics of the type of compounds, and for predicting senolytic activity.
L-135 In Figure 1, it is recommended to include selection criteria for narrowing down the candidates to be evaluated in the work. If the work is in a workflow diagram, the results should not be included. If it is a diagram with results, the selected compounds, or at least their number, should be presented at the end.
L-168 In Figure 2, include descriptions of sections A, B, and so on in the figure caption.
L-186 Improve the quality of Figures 3c and 3d.
L-234 Correct the reaction type to impersonal.
L-296 and L-327: What was the criterion for inclusion in this table for the 10/20 best compounds?
L-338: Reference 17 does not fully support the four targets presented. Include better bibliographic support.
L-441 Include the structure of metformin to analyze the structural comparison.
L-444 Delete the phrase "in this study" and focus on a more structured order.
L-484 Make the selection and cross-section of the 714 molecules obtained less clear, in order to focus on better screening and justify the best molecules evaluated.
L-620 Include a better definition of the optimization method for the reproducibility of the technique.
L-651 Include a reference to the metric evaluation method.
L-779 The work lacks a solid conclusion integrating all the results, as well as the lack of potential of the designed method.
Author Response
Reviewer 4 comments and response
Comments 1: The manuscript presents a comprehensive critical analysis of the application of a predictor for the discovery of new compounds and secondary metabolites for novel senolytic agents. The study covers everything from selection to the effect of senescence on the IMR-90 cell model, allowing for confirmation of the prediction model and virtual screening. However, attention to the following points is recommended:
Response 1: Thank you for your positive feedback and for highlighting the key aspects of our work. Your comments and suggestions are helpful in improving our manuscript. We have carefully revised the manuscript based on the comments and the detailed responses are as follows.
Comments 2: L-23 Eliminate personal language such as "we" to present an impersonal presentation.
Response 2: Thank you very much for your valuable feedback. We have reviewed the manuscript and revised a number of personal languages including L-23 to present impersonal presentations. With your suggestions, the linguistic presentation of our manuscripts has become more standardized.
Comments 3: L-112 Include the working hypothesis. Although the objective of the work is presented based on the review in the introduction, guidelines could be provided for generating the working hypothesis, including the characteristics of the type of compounds, and for predicting senolytic activity.
Response 3: Thank you for your advice. We revised this section to more clearly describe the working hypothesis and the research component of this study. (lines 98-113)
Comments 4: L-135 In Figure 1, it is recommended to include selection criteria for narrowing down the candidates to be evaluated in the work. If the work is in a workflow diagram, the results should not be included. If it is a diagram with results, the selected compounds, or at least their number, should be presented at the end.
Response 4: Thank you for your careful suggestion. We modified Figure 1 to remove the results section. And then, we added the criteria for compound selection in the figure notes section.
Comments 5: L-168 In Figure 2, include descriptions of sections A, B, and so on in the figure caption.
Response 5: Thank you for your advice. We have added a more detailed description to the figure notes section of Figure 2 to make the figure notes more complete.
Comments 6: L-186 Improve the quality of Figures 3c and 3d.
Response 6: Thank you for your advice. We have modified the colors of Figures 3c and 3d to improve the quality.
Comments 7: L-234 Correct the reaction type to impersonal.
Response 7: Thank you for your careful suggestion. As with comment 2, we have reviewed the manuscript and revised a number of personal languages to present impersonal presentations.
Comments 8: L-296 and L-327: What was the criterion for inclusion in this table for the 10/20 best compounds?
Response 8: Thank you for your feedback. Our criteria for selecting the 10/20 best compounds were those top 10/20 compounds that had the highest sum of scores modeled by SVM and MLP. We added this description in the notes of Table 1, 2. (lines 255, 308)
Comments 9: L-338: Reference 17 does not fully support the four targets presented. Include better bibliographic support.
Response 9: Thank you for your careful advice. We had cited a review here, which may indeed be inappropriate. At your suggestion, we replaced it with four research papers to support the conclusions.
Comments 10: L-441 Include the structure of metformin to analyze the structural comparison.
Response 10: Thank you very much for your valuable feedback. We have added the structure information of Metformin in Table 4.
Comments 11: L-444 Delete the phrase "in this study" and focus on a more structured order.
Response 11: Thank you for your advice. We have modified the description (line 453)
Comments 12: L-484 Make the selection and cross-section of the 714 molecules obtained less clear, in order to focus on better screening and justify the best molecules evaluated.
Response 12: Thank you for your feedback. We have revised this section to make the presentation clearer. (lines 497-508)
Comments 13: L-620 Include a better definition of the optimization method for the reproducibility of the technique.
Response 13: Thank you for your constructive feedback. As suggested, we added more detailed parameters and definitions for building CNN models.(lines 643-649)
Comments 14: L-651 Include a reference to the metric evaluation method.
Response 14: Thank you for your advice. We have added a reference to the metric evaluation method in the revised manuscript.(lines 688)
Comments 15: L-779 The work lacks a solid conclusion integrating all the results, as well as the lack of potential of the designed method.
Response 15: Thank you for your feedback. In our revised manuscript, we have added a separate conclusions section to highlight the main methods and results of the paper. (lines 567-580)

Round 2
Reviewer 2 Report (New Reviewer)
Comments and Suggestions for Authors
The manuscript has been sufficiently improved.
Reviewer 3 Report (New Reviewer)
Comments and Suggestions for Authors
All the issues and concerns raised in the previous round of review have been addressed in the latest version of the manuscript through the necessary revisions. I believe the work is suitable for publication in its current form.
This manuscript is a resubmission of an earlier submission. The following is a list of the peer review reports and author responses from that submission.
Round 1
Reviewer 1 Report
Comments and Suggestions for Authors
In this study, authors present a powerful senolytic predictor built using phenotypic data and machine learning techniques to identify compounds with potential senolytic activity. They curated a comprehensive training dataset consisting of 111 positive and 3951 negative compounds sourced from the literature. The dataset was used to train machine learning models incorporating traditional molecular fingerprints, molecular descriptors, and MoLFormer molecular embeddings. By applying MoLFormer-based oversampling and testing different algorithms, we found that the Support Vector Machine (SVM) and Multilayer Perceptron (MLP) models with MoLFormer embeddings exhibited the best performance, achieving AUC scores of 0.998 and 0.997, and F1 scores of 0.948 and 0.941, respectively. They then used this senolytic predictor to perform virtual screening of compounds from the DrugBank and TCMbank databases.
In the future, it expect that further experimental validation and model optimization will drive more accurate predictions, thereby facilitating the early discovery of senolytics and reducing experimental costs.
The authors work in the field of development of senolytic compounds capable of cleaning senescent cells. Neural networks are used to predict such compounds, and in general, the method of selecting compounds is justified and innovative. The article is recommended for publication in its current form.
Author Response
Reviewer 1 comments and response
Comment 1: In this study, authors present a powerful senolytic predictor built using phenotypic data and machine learning techniques to identify compounds with potential senolytic activity. They curated a comprehensive training dataset consisting of 111 positive and 3951 negative compounds sourced from the literature. The dataset was used to train machine learning models incorporating traditional molecular fingerprints, molecular descriptors, and MoLFormer molecular embeddings. By applying MoLFormer-based oversampling and testing different algorithms, we found that the Support Vector Machine (SVM) and Multilayer Perceptron (MLP) models with MoLFormer embeddings exhibited the best performance, achieving AUC scores of 0.998 and 0.997, and F1 scores of 0.948 and 0.941, respectively. They then used this senolytic predictor to perform virtual screening of compounds from the DrugBank and TCMbank databases.
In the future, it expects that further experimental validation and model optimization will drive more accurate predictions, thereby facilitating the early discovery of senolytics and reducing experimental costs.
The authors work in the field of development of senolytic compounds capable of cleaning senescent cells. Neural networks are used to predict such compounds, and in general, the method of selecting compounds is justified and innovative. The article is recommended for publication in its current form.
Response 1: Thank you very much for the time and effort you have put into reviewing our work. We appreciate the recognition of the importance and relevance of our work. Your feedback reinforces the significance of our research and motivates us to continue our efforts in improving the model through further experimental validation and optimization.
Reviewer 2 Report
Comments and Suggestions for Authors
This manuscript developed SVM and MLP models with MoLFormer embeddings to predict senolytic structures. The highlights of this work are MoLFormer-based oversampling to address the imbalanced data issue and the success of applying the models to perform virtual screening of compounds from the DrugBank and TCMbank databases. Overall, this paper provides detailed description of methods, and the conclusions are supported by the results. I only have a few comments.
- For CNN model, I was wondering if the authors considered adjusting kernel size to improve its performance.
- For methods selection, I was wondering why the authors tried CNN instead of other types of neural networks such as graph neural networks which seem to handle molecular structure information better than CNN.
- Please increase the image size and resolution in Figure 2.
- Please improve the equation format in line 242.
Author Response
Reviewer 2 comments and response
Comments 1: This manuscript developed SVM and MLP models with MoLFormer embeddings to predict senolytic structures. The highlights of this work are MoLFormer-based oversampling to address the imbalanced data issue and the success of applying the models to perform virtual screening of compounds from the DrugBank and TCMbank databases. Overall, this paper provides detailed description of methods, and the conclusions are supported by the results. I only have a few comments.
Response 1: Thank you for your positive feedback and for highlighting the key aspects of our work, particularly the MoLFormer-based oversampling and the successful application of our models for virtual screening. We appreciate your acknowledgment of the detailed description of our methods and the support of our conclusions by the results. Your comments and suggestions are helpful in improving our manuscript. We have carefully revised the manuscript based on the comments and the detailed responses are as follows.
Comments 2: For CNN model, I was wondering if the authors considered adjusting kernel size to improve its performance.
Response 2:Thank you for your constructive feedback. Adjusting the kernel size can indeed affect the performance of the CNN model, and a more detailed evaluation is necessary. In the revised manuscript, we have provided an update based on your suggestion.
In Supplementary Figure 2a of the original manuscript, we described the CNN model, which used a kernel size of 3 for the single convolutional layer model. For the models with multiple convolutional layers, we employed a kernel size of 5 for the first layer, while subsequent layers used a kernel size of 3. Following your recommendation, we conducted a more thorough adjustment of the kernel sizes, and the results are presented in the new Supplementary Figure 3. The kernel sizes for the first convolutional layer in our CNN models with varying layers were set to 3, 5, 7, and 9, respectively. The performance of the model was slightly improved by tuning the convolutional kernel size. Using the F1 score as an evaluation metric, the model with the first convolutional kernel size of 7 achieved the best performance among models with one or two convolutional layers, and the model with the first convolutional kernel size of 5 achieved the best performance among models with three convolutional layers.
Additionally, we have updated the comparison between the CNN models and the MLP models. We selected the CNN model with the highest F1 score among those with different convolutional layers for comparison with the MLP model, as shown in the new Supplementary Figure 5a. As illustrated in the figure, while the performance of these models improves slightly, the results remain consistent with our original conclusions: none of the CNN models outperformed the MLP model, and the F1 scores tended to decrease as the number of convolutional layers increased.
Comments 3: For methods selection, I was wondering why the authors tried CNN instead of other types of neural networks such as graph neural networks which seem to handle molecular structure information better than CNN.
Response 3: Thank you very much for your valuable feedback. Following your recommendation, we have tried the message passing graph neural network model proposed by Wang et al. and the graph neural network developed by Tsubaki et al. In the revised manuscript, we have updated these results, and the performance of the model is shown in Supplementary Figure 2.
However, the performance of these models was not satisfactory. The MPNN model by Wang et al. uses the SMILES representation of molecules as input and employs the Rdkit package to calculate structural information. The model integrates local information contained in each atom and bond to produce a prediction score as output. The MPNN model achieved a commendable AUC value of 0.854; however, the precision and recall were relatively low, resulting in an F1 score of only 0.269. Similarly, the GNN model by Tsubaki et al. also uses SMILES as input, learns molecular structural representations through GNN, and then utilizes MLP for downstream prediction tasks. However, the overall performance of this model was poor, with an average AUC value of 0.671 and an auPRC value of only 0.165.
In addition, we have updated the references that are cited in the manuscript (lines 504-510):
Smer-Barreto V, Quintanilla A, Elliott RJR, Dawson JC, Sun J, Campa VM, Lorente-Macías Á, Unciti- Broceta A, Carragher NO, Acosta JC, Oyarzún DA. Discovery of senolytics using machine learning. Nat Commun. 2023 Jun 10;14(1):3445.
Tsubaki M, Tomii K, Sese J. Compound-protein interaction prediction with end-to-end learning of neural networks for graphs and sequences. Bioinformatics. 2019 Jan 15;35(2):309-318.
Comments 4: Please increase the image size and resolution in Figure 2.
Response 4: Thank you for your advice. As suggested, we have increased the image size and improved the overall quality of the images to ensure better visibility and clarity(line 166).
Comments 5: Please improve the equation format in line 242.
Response 5: Thank you for your careful suggestion. We have modified the font size and equation to be center-aligned in order to enhance readability(line 628).
Finally, thank you once again for your insightful comments, which have greatly contributed to improving our manuscript.
Reviewer 3 Report
Comments and Suggestions for Authors
This is computational study relaying on virtual screening with machine learning models. The authors use a combination of molecular fingerprints, molecular descriptors, and MoLFormer embeddings to train their models and predict potential senolytic compounds from two databases.
While your study presents a novel and computationally rigorous approach to predicting senolytic compounds using machine learning, the lack of experimental validation significantly limits its suitability for publication in Molecules. The journal typically expects a balance between computational predictions and experimental verification to ensure the biological relevance of the findings. Without experimental validation, the predictions remain theoretical and lack the necessary confirmation of their senolytic activity. We recommend incorporating experimental validation of at least a subset of the predicted compounds to strengthen the manuscript and align it with the journal's standards. Alternatively, this work may be well-suited for submission to a journal where the focus is on methodological advancements and predictive modeling.
Author Response
Reviewer 3 comments and response
Comment 1: This is computational study relaying on virtual screening with machine learning models. The authors use a combination of molecular fingerprints, molecular descriptors, and MoLFormer embeddings to train their models and predict potential senolytic compounds from two databases.
While your study presents a novel and computationally rigorous approach to predicting senolytic compounds using machine learning, the lack of experimental validation significantly limits its suitability for publication in Molecules. The journal typically expects a balance between computational predictions and experimental verification to ensure the biological relevance of the findings. Without experimental validation, the predictions remain theoretical and lack the necessary confirmation of their senolytic activity. We recommend incorporating experimental validation of at least a subset of the predicted compounds to strengthen the manuscript and align it with the journal's standards. Alternatively, this work may be well-suited for submission to a journal where the focus is on methodological advancements and predictive modeling.
Response 1: Thank you very much for your constructive feedback. Experimental verification is indeed crucial for virtual screening work. We have recently started experimental work in the hope of discovering new senolytics and further improving our model in the future. Since some compounds in the Drugbank and TCMbank databases are difficult to obtain, we prioritized the experimental validation of compounds that are readily available. We have obtained a limited number of positive results, some of which we have included in the revised manuscript (Table 4 and Figure 6).
As senolytics are a class of anti-aging drugs, we are particularly interested in discovering compounds that can extend the lifespan or healthspan of organisms. Therefore, in our preliminary experimental validation, we chose to use the C.elegans model for screening anti-aging drugs, a commonly used model organism. We performed a preliminary screening through C.elegans lifespan experiments. As shown in Table 4, the predicted active compound voclosporin markedly extended the lifespan of C.elegans, achieving a lifespan extension rate of 19.1%, which is better than that of the well-known anti-aging drug metformin (11.3%). The other four compounds (simeprevir, belinostat, paritaprevir, and tenapanor), which were predicted to be inactive, had no effect on extending the lifespan of C.elegans. Voclosporin, from the Drugbank dataset, is a known calcineurin inhibitor used for treating autoimmune diseases. To our knowledge, there are currently no anti-aging studies on voclosporin. The specific anti-aging mechanisms of voclosporin will require further experiments to elucidate.
Due to the early stage of our experimental process, many of the predicted active candidate compounds have yet to undergo preliminary experimental screening. We apologize for the limited amount of experimental results we can provide at this time.
Finally, thank you once again for your insightful suggestions. Your feedback has made us more aware of the importance of experimental validation in computational work. In the future, we will place greater emphasis on this issue to make our results more convincing.
Reviewer 4 Report
Comments and Suggestions for Authors
The theoretical article “Development and Application of a Senolytic Predictor for 2 Discovery of Novel Senolytic Compounds and Herbs” is dedicated to creating and applying an in silico predictor for senolytic drug development. This senolytic predictor was developed based on QSAR methods, using literature data as a training set, and was used for virtual screening to potential senolytics in the DrugBank and TCMbank databases.
Minor comments:
- In the section “1. Introduction,” the authors write: “Using this predictor, we screened the DrugBank 95 dataset to identify 130 potential senolytic compounds for drug repurposing.” - This statement, appropriate experimental studies using these 130 potential senolytic compounds are needed. Correct the sentence.
- The section "2.1. Construction of Senolytic and Screening Datasets for Machine Learning": using data from Drugbank database version 5.1.10 need reference:
Wishart DS, Feunang YD, Guo AC, Lo EJ, Marcu A, Grant JR, Sajed T, Johnson D, Li C, Sayeeda Z, Assempour N, Iynkkaran I, Liu Y, Maciejewski A, Gale N, Wilson A, Chin L, Cummings R, Le D, Pon A, Knox C, Wilson M. DrugBank 5.0: a major update to the DrugBank database for 2018. Nucleic Acids Res. 2017 Nov 8. doi: 10.1093/nar/gkx1037.
- The section “3.5. Identification of Structurally Novel Senolytic Candidates from the TCMbank Database”: It would be interesting and important for the article to conduct experimental studies in vitro for the most active 714 compounds and compare them with the predicted results.
- The authors write: “The top 10 ranked compounds are shown in Table 2. Among these, bufotalidin, helveticoside, cymarin, cymarol, and malayoside were identified as cardiac glycosides, a class previously associated with senolytic activity.” Provide references to support this.
- The section “2.7. Enrichment Analysis of TCMbank Prediction Results” - using materials from the KEGG Database, need to add a reference:
Minoru Kanehisa, Susumu Goto, KEGG: Kyoto Encyclopedia of Genes and Genomes, Nucleic Acids Research, Volume 28, Issue 1, 1 January 2000, Pages 27–30, https://doi.org/10.1093/nar/28.1.27
- Provide a web link for the RDKit package (v2023.9.4).
This article is distinguished by its accuracy and well-thought-out research strategy and leaves a positive impression.
Author Response
Reviewer 4 comments and response
Comments 1: In the section “1. Introduction,” the authors write: “Using this predictor, we screened the DrugBank 95 dataset to identify 130 potential senolytic compounds for drug repurposing.” - This statement, appropriate experimental studies using these 130 potential senolytic compounds are needed. Correct the sentence.
Response 1: Thank you for your feedback. Based on your suggestion, we have revised the sentence to read: “Using this predictor, we screened the DrugBank dataset and predicted 130 predicted senolytic compounds as drug repurposing candidates. These computational findings require experimental confirmation.” (lines 104-106). This change has made our statement more accurate.
Comments 2: The section "2.1. Construction of Senolytic and Screening Datasets for Machine Learning": using data from Drugbank database version 5.1.10 need reference:
Wishart DS, Feunang YD, Guo AC, Lo EJ, Marcu A, Grant JR, Sajed T, Johnson D, Li C, Sayeeda Z, Assempour N, Iynkkaran I, Liu Y, Maciejewski A, Gale N, Wilson A, Chin L, Cummings R, Le D, Pon A, Knox C, Wilson M. DrugBank 5.0: a major update to the DrugBank database for 2018. Nucleic Acids Res. 2017 Nov 8. doi: 10.1093/nar/gkx1037.
Response 2: Thank you for your thorough review of our manuscript. We have added the citation of DrugBank in the appropriate section (line 494). While we have referenced this paper in the Results section, we overlooked it in the Methods section. We appreciate your thoughtful suggestion.
Comments 3: The section “3.5. Identification of Structurally Novel Senolytic Candidates from the TCMbank Database”: It would be interesting and important for the article to conduct experimental studies in vitro for the most active 714 compounds and compare them with the predicted results.
Response 3: Thank you very much for your constructive feedback and suggestions. Experimental verification is indeed crucial for virtual screening work. We have recently started experimental work in the hope of discovering new senolytics and further improving our model in the future. Since some compounds in the Drugbank and TCMbank databases are difficult to obtain, we prioritized the experimental validation of compounds that are readily available. We have obtained a limited number of positive results, some of which we have included in the revised manuscript (Table 4 and Figure 6).
As senolytics are a class of anti-aging drugs, we are particularly interested in discovering compounds that can extend the lifespan or healthspan of organisms. Therefore, in our preliminary experimental validation, we chose to use the C.elegans model for screening anti-aging drugs, a commonly used model organism. We performed a preliminary screening through C.elegans lifespan experiments. As shown in Table 4, the predicted active compound voclosporin markedly extended the lifespan of C.elegans, achieving a lifespan extension rate of 19.1%, which is better than that of the well-known anti-aging drug metformin (11.3%). The other four compounds (simeprevir, belinostat, paritaprevir, and tenapanor), which were predicted to be inactive, had no effect on extending the lifespan of C.elegans. Voclosporin, from the Drugbank dataset, is a known calcineurin inhibitor used for treating autoimmune diseases. To our knowledge, there are currently no anti-aging studies on voclosporin. The specific anti-aging mechanisms of voclosporin will require further experiments to elucidate.
Due to the early stage of our experimental process, many of the predicted active candidate compounds have yet to undergo preliminary experimental screening. We apologize for the limited amount of experimental results we can provide at this time.
Finally, thank you once again for your insightful suggestions. Your feedback has made us more aware of the importance of experimental validation in computational work. In the future, we will place greater emphasis on this issue to make our results more convincing.
Comments 4: The authors write: “The top 10 ranked compounds are shown in Table 2. Among these, bufotalidin, helveticoside, cymarin, cymarol, and malayoside were identified as cardiac glycosides, a class previously associated with senolytic activity.” Provide references to support this.
Response 4: Thank you for your valuable feedback. Based on your suggestion, we have added the relevant references in that section (line 277).
(1) Guerrero A, Herranz N, Sun B, Wagner V, Gallage S, Guiho R, Wolter K, Pombo J, Irvine EE, Innes AJ, Birch J, Glegola J, Manshaei S, Heide D, Dharmalingam G, Harbig J, Olona A, Behmoaras J, Dauch D, Uren AG, Zender L, Vernia S, Martínez-Barbera JP, Heikenwalder M, Withers DJ, Gil J. Cardiac glycosides are broad-spectrum senolytics. Nat Metab. 2019 Nov;1(11):1074-1088.
(2) Triana-Martínez F, Picallos-Rabina P, Da Silva-Álvarez S, Pietrocola F, Llanos S, Rodilla V, Soprano E, Pedrosa P, Ferreirós A, Barradas M, Hernández-González F, Lalinde M, Prats N, Bernadó C, González P, Gómez M, Ikonomopoulou MP, Fernández-Marcos PJ, García-Caballero T, Del Pino P, Arribas J, Vidal A, González-Barcia M, Serrano M, Loza MI, Domínguez E, Collado M. Identification and characterization of Cardiac Glycosides as senolytic compounds. Nat Commun. 2019 Oct 21;10(1):4731.
Comments 5: The section “2.7. Enrichment Analysis of TCMbank Prediction Results” - using materials from the KEGG Database, need to add a reference:
Minoru Kanehisa, Susumu Goto, KEGG: Kyoto Encyclopedia of Genes and Genomes, Nucleic Acids Research, Volume 28, Issue 1, 1 January 2000, Pages 27–30, https://doi.org/10.1093/nar/28.1.27
Response 5: Thank you for your careful review and suggestion. Following your suggestion, we have added the references in the revised manuscript (line 612).
Kanehisa M, Goto S. KEGG: kyoto encyclopedia of genes and genomes. Nucleic Acids Res. 2000 Jan 1;28(1):27-30.
Comments 6: Provide a web link for the RDKit package (v2023.9.4).
Response 6: Thank you for your comment. Regarding the link to the RDKit package (v2023.9.4) that you mentioned, we have added the link in the revised manuscript (line 489).
Finally, thank you once again for your insightful comments, which have greatly contributed to improving our manuscript.
Round 2
Reviewer 3 Report
Comments and Suggestions for Authors
While I appreciate the improvements in the computational methodology and the addition of preliminary C. elegans data for voclosporin, I still find the manuscript lacking sufficient experimental validation to support the predicted senolytic compounds. The new data, though a step forward, is limited and does not provide mechanistic insights into senolytic activity. Without broader experimental validation, the study remains theoretical and is better suited for a purely computational journal. I recommend expanding experimental validation to strengthen the manuscript for publication in Molecules.